# Patterns and drivers of Holocene moisture variability in mid-latitude eastern North America

J. Sakari Salonen [1] ✉, Frederik Schenk [1,2,3], John W. Williams [4], Bryan Shuman [5], Ana L. Lindroth Dauner [1], Sebastian Wagner[6], Johann Jungclaus[7], Qiong Zhang [3,8] & Miska Luoto [1]

Proxy data for eastern North American hydroclimate indicate strong and persistent multi-millennial droughts during the Holocene, but climate model simulations often fail to reproduce the proxy-inferred droughts. Diagnosing the data–model mismatch can offer valuable insights about the drivers of hydrological variability and different regional sensitivities to hydroclimate forcing. Here we present a proxy–modeling synthesis for Holocene climates in the eastern North American mid-latitudes, including machine-learning-based water balance reconstructions and high-resolution climate simulations. These data-model results resolve prior-generation inconsistencies, show consistent spatiotemporal patterns of Holocene hydroclimate change, and enable assessment of the driving mechanisms. This agreement suggests that the secular summer insolation trend, combined with the Laurentide Ice Sheet deglaciation and its effect on atmospheric circulation, together explain the extent and duration of drier-than-present climates. In addition, our high-resolution proxy data and transient simulations reveal clear multi-centennial climate variability. In our simulations, temperature-driven increases in eva-potranspiration exceed regional precipitation gains, drying much of the region during the mid Holocene. This suggests that the mid-Holocene multi-millennial drought was driven by similar processes compared to the drying trajectory projected for mid-latitude North America over this century, which is also primarily driven by warming.

Changes in water availability have a major impact on both global- and regional-scale ecosystem processes[1,2] and societies[3]. In the North American mid-continent, climate model projections for the 21st century indicate a trend towards increased atmospheric and hydrological aridity with rising greenhouse gas emissions[4,5]. Regional water scarcity combined with diminishing groundwater reserves mean that both wildland and agrarian ecosystems are sensitive to changes in water availability[6,7]. Because the recent decades of anthropogenic warming have already pushed the global climate system beyond the limits of historical record[5], paleoclimate archives can offer valuable insights

[1]Department of Geosciences and Geography, University of Helsinki, Helsinki, Finland. [2]Department of Geological Sciences, Stockholm University, Stockholm, Sweden. [3]Bolin Centre for Climate Research, Stockholm University, Stockholm, Sweden. [4]Department of Geography and Center for Climatic Research, University of Wisconsin-Madison, Madison, WI, USA. [5]Department of Geology and Geophysics, University of Wyoming, Laramie, WY, USA. [6]Institute for Coastal Research – Analysis and Modeling, Helmholtz-Zentrum Hereon, Geesthacht, Germany. [7]Max-Planck-Institute for Meteorology, Hamburg, Germany. [8]Department of Physical Geography, Stockholm University, Stockholm, Sweden. ✉e-mail: sakari.salonen@helsinki.fi

into the drivers of hydrological variability and regional implications of long-term drying trends[8]. Proxy data for past moisture variations suggest that the 20th- and 21st-century hydroclimate regime in North America is unusually wet relative to the Holocene, with most of the Holocene characterized by drier-than-present conditions for most of the United States (except southwest) and southern Canada[9]. Current climate models poorly simulate these Holocene moisture variations, e.g. showing greater-than-present precipitation in mid-latitude North America during the early-to-mid Holocene, a stark contrast to the peak aridity suggested by the proxy data[9,10] and underpredicting the magnitude of mid-Holocene lake-level drawdown in the mid-continent[11]. Moreover, many geomorphic and ecological systems show high climatic sensitivity and non-linear threshold responses to the modest forcings over the Holocene, with wide variations among sites in the timing and pace of drying, suggesting the possibility of unpredictable and potentially abrupt local responses to regional aridification[8].

Late-Quaternary hydroclimatic variations in eastern North America have been inferred from many sources[8,9,12,13], including fossil pollen, eolian deposits (dune mobilization, loess deposition), carbon isotopes (abundance of $C_4$ vs. $C_3$ grasses[6]), testate amoebae[14], indicators of lake water balance such as changes in physical sedimentology[1,12,15], oxygen-isotope and calcite-aragonite ratios[6], and diatom indicators of salinity[16]. Among these proxies, fossil pollen records have been a backbone of Holocene paleohydrological reconstructions in central and eastern North America[17-20], because of the widespread availability of lake archives, a demonstrated sensitivity of vegetation composition (and consequently, pollen assemblages) to variations in water availability and temperature, and orthogonal regional gradients in temperature and moisture. The pollen-based quantitative paleoclimatic reconstructions typically used methods such as the modern analog technique (MAT[12,21]), response surfaces (a smoothed form of MAT[18]), or other classical transfer functions such as weighted averaging-partial least squares[13,22]. Other studies have employed semi-quantitative paleoclimate measures such as z-scores of proxy variability[9], or quantitative reconstructions of variables indirectly related to moisture balance, including salinity[23] or changes in vegetation cover[24].

Key features of prior paleoclimatic reconstructions include long-term increases in moisture availability that produced wetter conditions today across much of mid-latitude North America compared to the early- and mid-Holocene, as well as abrupt millennial and centennial departures from these trends. Regionally, while the mid-continent was likely wet during the early Holocene, the eastern coastal areas were substantially drier than today[25]. The pattern reversed around ca. 8 ka, when large portions of the mid-continent dried[8,15,19,23,26], in contrast to the eastern areas that experienced ~20% increases in effective precipitation[1,27,28]. A leading hypothesis suggests that the hydroclimate reorganization reflects the rapidly diminishing influence of the Laurentide Ice Sheet towards the end of the early Holocene[12,15]. After around 8 ka, a long-term increase in effective moisture, amplified by a rapid moistening at around 5.5 ka, affected most areas, possibly in response to long-term insolation trends and superimposed millennial variability[12,29]. These millennial and multi-centennial hydroclimate influenced eolian activity in the Great Plains[30,31], altered water levels in lakes and wetlands[14,32], and transformed forest composition[1].

In this study, we present an analysis and synthesis of hydroclimate trends in the eastern North America, drawing on recent advances in the available proxy data, quantitative climate reconstruction methodology, and paleoclimate simulations. First, machine-learning based approaches have been adopted in proxy-based paleoclimatic reconstructions. These include ensemble models of regression trees (e.g., random forests, boosted regression trees) and artificial neural networks[33]. Here we focus on regression tree ensembles, which were first applied in biogeography to contemporary ecosystems[34,35], and are now increasingly applied to paleoclimatic reconstruction (reviewed in refs. 33,36,37). A key strength of regression-tree based models is their

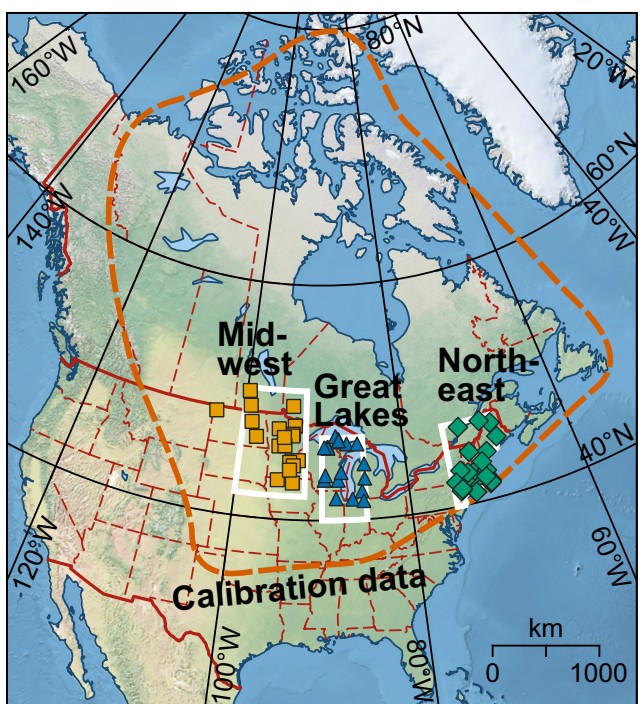

**Fig. 1 | Proxy data sites.** Fossil pollen records used (for details and references, see Supplementary Table 1) are grouped in three clusters: Midwest (*squares*), Great Lakes (*triangles*), and Northeast (*diamonds*). The corresponding regions used to extract results from climate model simulations are indicated with *white lines*. The *dashed line* indicates the geographic span of the pollen–climate calibration dataset[55] (2419 surface pollen samples).

ability to detect comparatively weak climatic signals in datasets, such as those associated with secondary (or tertiary, etc.) environmental drivers[34]. Regression trees achieve this by first screening the training data for indicator taxa showing strong responses to each climatic variable, and then selectively focusing on these taxa in prediction[36,37]. This characteristic of regression trees is useful in moisture reconstruction because North American pollen datasets primarily respond to temperature (with summer temperature carrying more predictive power than winter temperature), but with a clear secondary signal of moisture, with certain pollen taxa responding strongly to moisture-related variables[17,20,37,38]. Second, a suite of high-resolution and well-dated pollen records from the last two decades (Supplementary Table 1) now permit assessment of centennial-scale climate variability at local to regional scales. Third, an increasing number of transient Earth system model simulations for the Holocene is now available[39], with higher spatial model resolution and various forcings, including solar and volcanic forcing[40] and dynamic vegetation changes[41] in addition to changes in orbital and greenhouse gas forcing.

Here we apply the boosted regression tree (BRT)-based proxy–climate calibration models[37] to prepare synthesis reconstructions of annual water balance (calculated as annual precipitation minus evapotranspiration[42]) and July mean temperature ($T_{jul}$). These reconstructions are then integrated with an ensemble of four climate model simulations for the Holocene. Our paleoclimate reconstructions are based on fossil pollen sequences from three regions spanning the eastern North American mid-latitudes: Midwest (MW), Great Lakes (GL) and Northeast (NE), which collectively extend from the prairie–forest ecotone to the eastern seaboard (Fig. 1). While the general timing of multi-millennial drought is well established by available multi-proxy evidence, we explore open questions that especially require quantitative moisture reconstructions. First, we assess the magnitude and spatio-temporal progress of Holocene droughts, facilitating a direct

comparison with trends from the Holocene simulations. Second, using a high-resolution subset of the data, we examine the shorter, centennial-scale moisture variations, including their prevalence, geographic distribution, periodicities, and magnitudes. Finally, we test the hypothesis that the spatiotemporal patterns of Holocene aridification in eastern North America are linked to atmospheric circulation changes driven by the retreat of the Laurentide Ice Sheet (LIS).

## Results and Discussion
### Reconstructed millennial trends in moisture and temperature
Water balance reconstructions using the BRT calibration models from all three regions (Fig. 2a) show multi-millennial, early-to-mid Holocene

relative drought with virtually the entire error margins of the synthesis curves under the preindustrial value. There is a notable longitudinal gradient in the timing of peak drought, which commences first in the NE at interglacial onset, with a sharp Holocene water balance minimum of ca. −180 mm reached at around 11 ka. Conversely, in the MW, peak aridity is reached later in mid Holocene (ca. 7 ka), with a reduced amplitude of ca. −110 mm. In the GL, the timing of the the water balance minimum at 10−6 ka falls between those in NE and MW and has a similar amplitude with MW at −100 mm. The fossil pollen samples generally have good modern analogs, with the compositional distance (squared-chord distance) to best-matching modern pollen sample largely around 0.1−0.15, and only 288 fossil samples out of 5257 (5.5%)

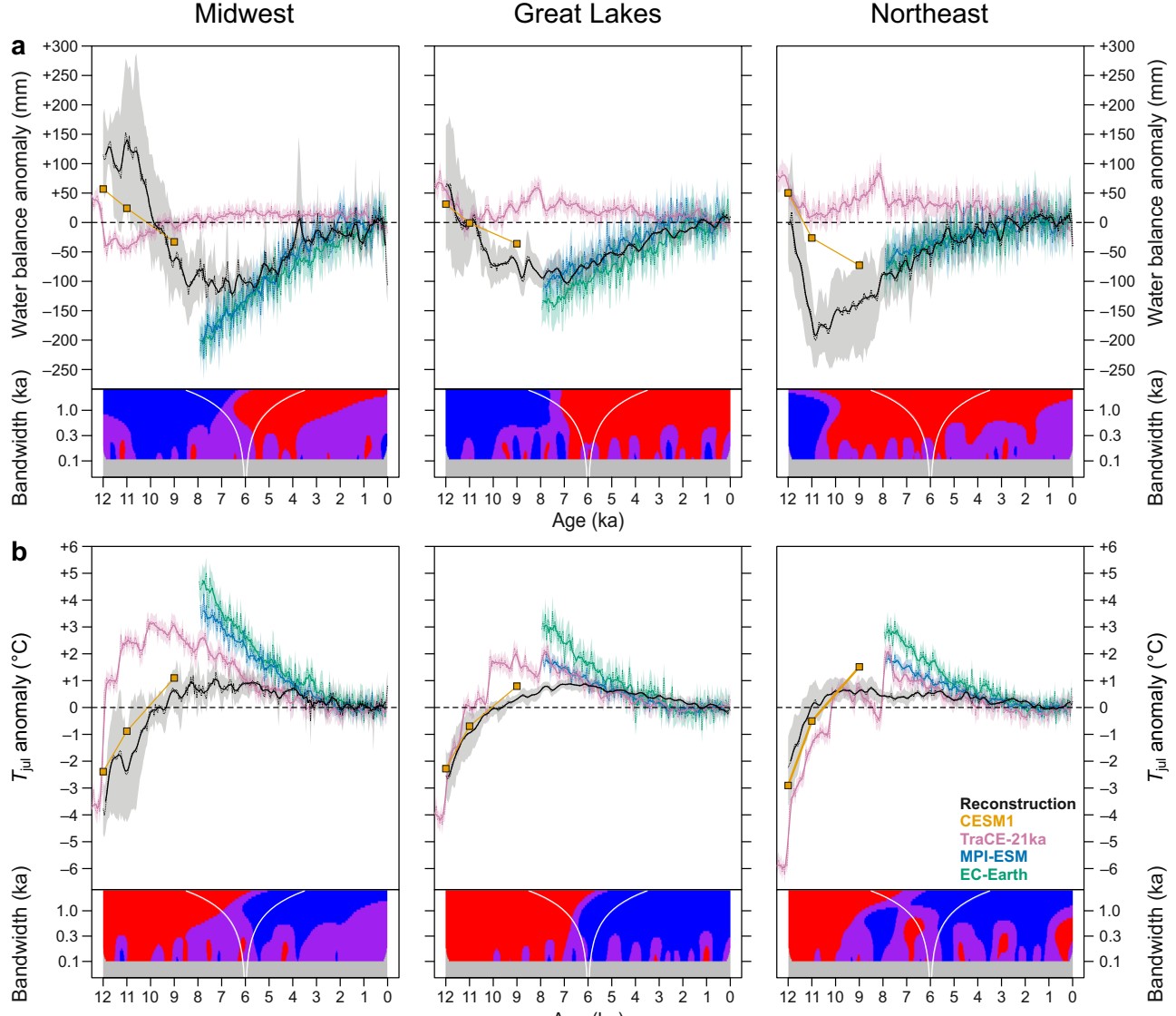

**Fig. 2 | Climate anomalies in the reconstructions and model simulations.** Comparison of reconstructed and simulated anomalies is shown for (**a**) annual water balance and (**b**) July mean temperature ($T_{jul}$) for the Midwest, Great Lakes and Northeast regions. Reconstructions from fossil pollen sequences are expressed as the mean of all reconstructions interpolated at 50-year time step (*dotted line*) and a five-point running mean (*solid line*). The uncertainty bands for the reconstructions represent 95% errors of the ensemble mean calculated using 1000 bootstrap samples of all datasets. Three transient climate simulations are shown, including TraCE-21ka (past 12 ka), MPI-ESM (past 8 ka), and EC-Earth (past 8 ka), with *dotted lines* indicating the means of annual values calculated for 50-year bins and the *solid lines* the five-point averages of the bin means. The 95% error bands for the transient simulations were calculated as ± 2σ of the values in each 50-year bin divided by the

square root of the sample size (50). *Squares* indicate anomalies from CESM1 equilibrium simulations for 12, 11, and 9 ka. The anomalies are expressed relative to the preindustrial period (0.25−0.75 ka) for the reconstructions and the transient simulations, and relative to a preindustrial control run (Supplementary Table 2) for CESM1. The SiZer maps (lower panels) show the significant features of the BRT-based reconstructions when smoothed at a range of bandwidths, with the bandwidth used at each point on the vertical axis indicated by the horizontal distance between the *white lines*. For each point in time and each bandwidth (h), *red* indicates a significant rising trend, *blue* a significant falling trend, *purple* a lack of a significant trend, and *gray* a lack of sufficient data for meaningful inference. For reconstructions from the individual fossil data sites, see Supplementary Figs. 1–10.

exceeding a threshold distance of 0.25 recommended for a good analog[38] (Supplementary Figs. 1–10). In the MW and NE, the analog quality remains good through the Holocene suggesting a lack of biases in the reconstructions due to poor modern analogs. However, in the GL analog quality is reduced through 12–10 ka (Supplementary Fig. 5). In consequence, in the GL, the timing of the onset of negative water balance anomalies is left uncertain, as the wide error band allows for trajectories ranging from an NE-like early onset to one identical to MW. The $T_{jul}$ reconstructions (Fig. 2b) show a Holocene temperature maximum (HTM) with a broadly uniform timing in all regions, with the millennial-bandwidth smoothers indicating a plateau at ca. 7 ka (Fig. 2b, lower panels), however with the error bands in the NE allowing for the possibility of an earlier temperature maximum at 11–8 ka. The amplitude of peak warmth decreases towards the Atlantic (ca. 0.9–1.0 °C in the MW, 0.8–0.9 °C in the GL, and 0.6–0.7 °C in the NE).

Some aspects of the proxy reconstructions largely confirm prior findings about the broadscale millennial moisture and temperature trends. In the well-studied MW region, the temporal evolution of the multi-millennial Holocene drought matches that identified in previous studies based on shifts in pollen percentages of forbs and moisture-sensitive tree types, corresponding pollen-based precipitation reconstructions, and lake-level reconstructions, with drying commencing at ca. 10 ka and with peak drought conditions reached by ca. 7 ka[15,17,19,20] (Fig. 2a). In contrast, NE shows an early-Holocene drought followed by an increasing water balance and multi-century variability since 11 ka (Fig. 2a), which aligns with recent lake-level and pollen data[15,32] and derived quantitative moisture reconstructions from this region[1,20,29]. However, fewer prior data exist for the intervening GL sector, where our water balance reconstruction shows that the timing of the lowest water balance in the early Holocene occurs between the timing in MW and NE (Fig. 2a).

The pacing of reconstructed water balance vs. $T_{jul}$ changes is roughly synchronous in MW, while in GL and NE, the earlier onset of drought results in the water balance minimum preceding the temperature maximum (Fig. 2a,b), supporting the notion that independent signals of $T_{jul}$ and water balance can be extracted from eastern North American fossil pollen data[20,37,38]. Our $T_{jul}$ reconstructions could be slightly biased by postglacial vertical land movements (forebulge collapse), with subsidence in the southern part of the GL region estimated at close to 100 m since 10 ka[43]. At present, the rate of subsidence is highest in the MW and GL, while for NE the rate is smaller or transitions to glacio-isostatic uplift in the northern part of this cluster[44]. The climatic impact of the vertical land movements could be regionally up to ~ 0.6 °C based on a lapse rate of 6.4 °C/km, which falls within the uncertainty intervals of our reconstructions. However, the expected effects of the subsidence are to negatively bias the HTM amplitudes, due to higher mid-Holocene site elevation in forebulge collapse areas, and to delay the reconstructed timing of the HTM because the negative temperature bias increases towards early Holocene and thus delays the arrival to the mid-Holocene temperature maximum. Thus the bias would be strongest in the mid-continent and so would tend to mute the reconstructed gradient in HTM amplitude, which in our reconstructions increases from the Atlantic towards the mid-continent. Hence, correcting for the glacio-isostatic effect would reinforce the observed pattern.

A principal components analysis (PCA) of the water balance reconstructions covering all three site clusters reveals two dominant modes of variability (Fig. 3a), with the first component (PC1) explaining 37.0% and the second component (PC2) 22.5% of the total variance, and the remaining components not exceeding 6.1%. As shown by the positive PC1 site loadings at nearly all sites (50 out of 53; Fig. 3b), PC1 captures the significant shared water balance trend of all sites with an early-to-mid-Holocene drought transitioning into a late-Holocene wetting pattern. This trend is evident in PC1 sample

scores, which show a decline until 8 ka, followed by a rising trend towards the present (Fig. 3c). Based on the PC2 site loadings (Fig. 3d), showing clusters of positive loadings in MW and negative loadings in NE, PC2 represents the temporal variation in longitudinal water balance contrast. The PC2 sample scores (Fig. 3e) reach a maximum in the Early Holocene (ca. > 8.5 ka), representing the period of maximum contrast in water balance anomalies due to peak multi-millennial drought in NE before the early wet conditions ended in MW (Fig. 2a). However, the northernmost sites in NE showing positive PC2 loadings (Fig. 3d) do not reflect this trend, instead covarying with the mid-continental sites.

## Comparison with climate model simulations

Overall, the simulations with CESM1, MPI-ESM and EC-Earth agree well with the evolution of water balance for the past 12 ka as reconstructed with our BRT-based calibration models (Fig. 2a). This agreement signifies an improvement over earlier simulations by TraCE-21ka and gives confidence in both the reconstructions and the numerical models. The initial wet conditions during the deglaciation period (12 to 11 ka) in the MW and GL region, and the following drying trend towards 9 ka in reconstructions is well captured by the CESM1 snapshot simulations. Conversely, the transient simulation with TraCE-21ka falls largely outside the uncertainty band of reconstructions throughout the entire period, with almost no changes in water balance.

The long-term wetting trend over the past 8 ka is well captured by MPI-ESM and EC-Earth simulations, separating these models from the more inconsistent hydrological trends presented by TraCE-21ka for the Holocene. Despite these advancements, some data–model discrepancies persist, such as the underestimation of early-Holocene dry conditions in the NE by CESM1, which may result from a poor spatial coverage of coastal areas in the model. Meanwhile, MPI-ESM and EC-Earth simulate slightly drier conditions than reconstructed from 8–7 ka for the MW, resulting in a stronger wetting trend than seen in the reconstructions.

The CESM1 simulations also show better agreement with the proxy reconstructions for $T_{jul}$ than the transient simulations from TraCE-21k, MPI-ESM, and EC-Earth (Fig. 2b). While the MPI-ESM and EC-Earth simulations overcome the previous noted inconsistency in water balance seen in TraCE-21k, their simulated $T_{jul}$ trends for the last 8 ka are significantly higher than in the reconstructions with up to +4.5 °C in the MW for EC-Earth and +3.5 °C for MPI-ESM) (Fig. 2b). Such deviations underscore the challenge of accurately reproducing $T_{jul}$ trends in the transient simulations.

Sensitivity of vegetation and pollen to changes in water balance grounds our reconstruction, but the simulations enable us to disentangle the changes in precipitation versus evapotranspiration, which the proxy record cannot separately resolve. Figure 4 shows the simulated anomalies in annual precipitation ($\Delta P$) and potential evapotranspiration ($\Delta PET$) relative to preindustrial levels, which together inform the observed changes in water balance changes ($\Delta[P–PET]$), which appear broadly consistent with the reconstructions in Fig. 2. Based on these hydrological anomalies, the wet conditions at 12 ka and 11 ka resulted primarily from reduced PET due to cooler and shorter than preindustrial summers, which compensated for a strong reduction in precipitation during this period (Fig. 4a). As the early Holocene climate warmed, however, the persistence of low precipitation favored drought conditions from 11 ka onward. For the mid-Holocene ~8 to 5 ka (Fig. 4b), the most severe drought conditions were driven by high PET during the substantially warmer than preindustrial summers. Conversely, the onset of neoglacial cooling and increased precipitation around 4 ka alleviated the drought conditions and allowed water balance to increase by 2 ka. It is noteworthy that the shift in PET from negative values before 8 ka to positive, but declining, afterwards predominantly shaped the evolution of the annual water balance curves in

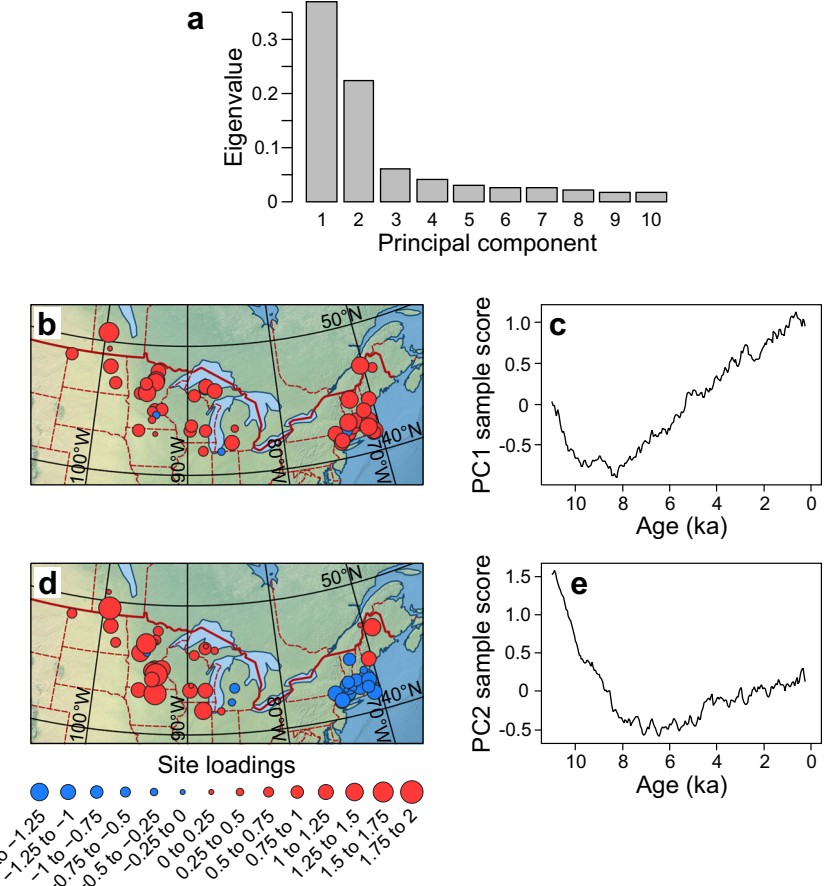

**Fig. 3 | Principal components (PC) analysis of the water balance reconstructions for 11 to 0.25 ka. a** Eigenvalues of the first ten principal components (PC). **b**–**e** Loadings for the reconstruction sites for PC1 (**b**) and PC2 (**d**). Sample scores of water balance reconstructions plotted across time for PC1 (**c**) and PC2 (**e**), where each sample comprises a set of reconstructed water-balance values from 53 sites for a single time point.

our simulations, as the transient dynamics in PET overprinted the roughly linear increase of annual precipitation from 12 to 4 ka. This pattern underscores the dominant influence of PET, driven by thermal responses to ice sheet configuration and orbital forcing, on the regional water balance.

To investigate the large $T_{jul}$ difference in simulations for the mid-Holocene, we also examine the data–model agreement for growing degree days (GDD), a measure integrating the biologically available heat across the growing season. For GDD5 (5 °C temperature threshold), transient simulations show a good agreement with reconstructions for the past 8 ka in GL and NE, while MPI-ESM and EC-Earth are again too warm over the MW (Supplementary Fig. 11). In contrast to the good agreement with $T_{jul}$, CESM1 performs worse for GDD5, indicating low GDD5 values for 12 to 9 ka, albeit following a comparable trend with reconstructions. There is an apparent paradox in the models showing consistent water balance, despite indicating much higher $T_{jul}$ compared to proxies which would suggest a significant disagreement in the PET portion of the water balance equation. By design, water balance includes compensating effects between PET (driven by warm-season temperatures above freezing) and the amount of annual precipitation. While all components are known in climate models including the warm season length contributing to PET, no direct proxy evidence is available for PET and annual $P$, making the paradox difficult to fully resolve. However, for GDD5, which characterizes the overall warm season instead of $T_{jul}$ only, the agreement between proxies and the MPI-ESM and EC-Earth simulations improves considerably (Supplementary Fig. 11), which increases our confidence in the realism of the PET and precipitation contributions to the water balance of the

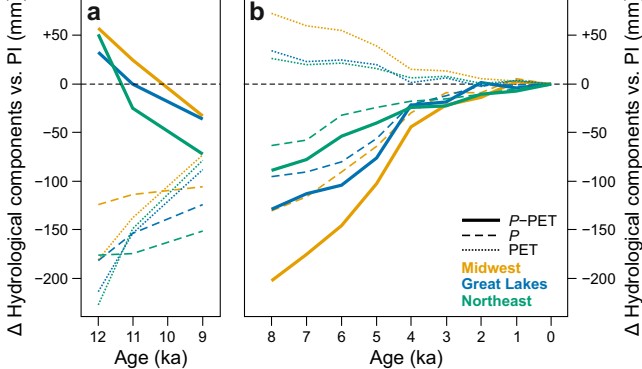

**Fig. 4 | Simulated contributions of precipitation and evapotranspiration to water balance.** Simulated evolution of hydrological variables that govern water balance changes is shown for the past 12 ka relative to preindustrial (PI) for the Midwest, Great Lakes, and Northeast regions ($P$ = precipitation, PET = potential evapotranspiration, $P$ − PET = water balance). The 12, 11, and 9 ka snapshots (**a**) are based on CESM1 simulations, while the past 8 ka (**b**) represent the average of MPI-ESM and EC-Earth transient runs. For the numerical paleoclimate anomalies from the models, see Supplementary Table 2.

past 8 ka indicated by MPI-ESM and EC-Earth (Fig. 4b). We hence consider it more likely that the water balance in the proxies and the models are correct and rather doubt the accuracy of the $T_{jul}$ inferences. This also applies to CESM1, with the difference that here $T_{jul}$ appears more similar with proxies than the warm-season quantity GDD5.

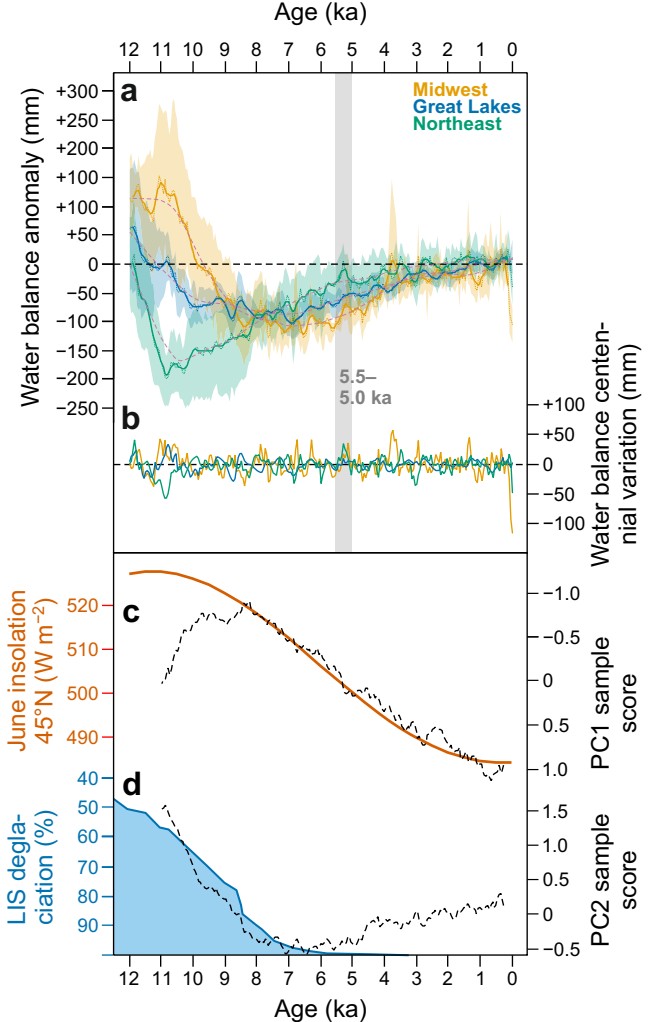

**Fig. 5 | Comparisons of water balance signals and supporting data. a** Water balance synthesis curves (*dotted lines*; five-point running mean is indicated with a *solid line*) with 95% error margins (*shaded bands*) for the Midwest, Great Lakes, and Northeast regions, expressed as deviations from the preindustrial (0.25–0.75 ka) mean. The *gray bar* indicates the centennial event at 5.5–5.0 ka. **b** Centennial-scale variations in the Midwest, Great Lakes, and Northeast water-balance reconstructions, calculated as the residuals of LOESS smoothers with a 2-ka span (*dashed lines* in panel **a**). **c–d** Regional forcing factors including June insolation at 45°N[78] (**c**) and Laurentide ice sheet (LIS) extent during deglaciation[79] (**d**). The insolation values in panel (**c**) are overlain by sample values for PC1 of the water balance reconstructions, and the LIS extent in (**d**) is overlain by PC2 (*dashed lines*).

## Sub-millennial climate events and periodicities

Drawing on the subset of well-dated and high-resolution fossil pollen records available for the MW and NE regions, our reconstructions confirm the presence of submillennial climate events and periodicities in temperature and moisture levels superimposed on the multi-millennial climate trends. At centennial smoothing bandwidths, both water balance and $T_{jul}$ in all regions show significant variations (lower panels in Fig. 2a, b) which are, however, generally not coherent among regions (Fig. 5a). Also, the amplitude of the centennial shifts (up to ca. 50 mm; Fig. 5b) is smaller than that of the multi-millennial aridity trend. The outstanding, spatially coherent centennial event occurs at ca. 5.5–5.0 ka, initiating with a significant centennial-scale rise in water balance in all regions at ca. 5.5 ka (Fig. 2a) and leading to a transient multi-century moisture maximum (Fig. 5a). While we cannot rule out that this is only a coincidental alignment of the continuous submillennial variations seen in all clusters, the 5.5–5.0 ka event matches a

wet period at 5.4 to 4.8 ka previously reported in the multiproxy data from the varved sequence of Elk Lake, Minnesota[19], which interrupts the mid-Holocene period of aridity. Shuman[29] also identified a period of increased moisture at 5.6–4.5 ka spanning the eastern North American mid-latitudes as the outstanding centennial-millennial scale deviation from the Holocene multi-millennial trends in this region. The amplitude of centennial moisture changes is overall stable through the Holocene in all regions (Fig. 5b). The one exception is a sharp apparent reduction in water balance in MW over the past 200 years (Fig. 5a), which is likely a spurious effect driven by the concurrent and probably anthropogenic upswings in *Artemisia* and Amaranthaceae pollen percentages (Supplementary Figs. 12–13).

Numerous significant periodicities in water balance and $T_{jul}$ were found in wavelet analyzes, which we ran on both the transient simulations and a subset of proxy reconstructions based on high-resolution pollen sequences available in the MW and NE clusters. These results are summarized in Fig. 6 as kernel density estimates fitted to the detected significant periodicities in the reconstructions (Fig. 6a, b) and the simulations (Fig. 6c, d). The periodicities are broadly similar for water balance (dotted lines) and $T_{jul}$ (dashed lines), suggesting the influence of summer temperature variations on moisture dynamics, even at sub-millennial timescales (compare with Fig. 4). A notable clustering of periodicities around 0.2 ka is observed for both reconstructions and simulations in both MW and NE. However, the NE reconstructions also show a larger cluster between 0.4 and 0.5 ka, a feature not captured in the simulations. On the other hand, the simulations reveal significant multidecadal periodicities, which cannot be evaluated in the reconstructions given the insufficient temporal sampling of the fossil pollen datasets.

The ~200-year periodicities are also found in earlier simulation experiments with and without solar forcing, and may either reflect the role of solar forcing on North American hydroclimates, since cycles around 210 years has been linked to Suess/de Vries cycles, or represent internal variability[39,45,46]. In control simulations with the same EC-Earth model as used here for the past 8 ka, internal feedbacks are enough to sustain multi-centennial variability of the Atlantic meridional overturning circulation[47] without any changes in external forcing. Simulated and reconstructed low-frequency variability is hence likely the result of combined internally and externally forced variability. The diminishing series of peaks at 0.5–1.2 ka periodicities in the MW proxy data (Fig. 6a) may represent subharmonics of the 200-year period. In NE reconstructions, however, the significant periodicities cluster around 400 and 500 years, especially for $T_{jul}$ (Fig. 6b). Similar periodicities were observed in the marine sediment cores retrieved from the North Atlantic and Arctic oceans[48–50] and also in earlier analyzes of lake-level reconstructions as well as the pollen data in the NE region[29]. Because the lakes of the NE cluster are located closer to the coastline, these periodicities (400–500 years) might be the effect of variations in ocean circulation on top of any solar forcing signal[51], a feature that is not captured by proxies in the more continental MW region. An alternative explanation for the comparatively weaker 200-year periodicity in NE is the lower average resolution in NE fossil pollen sequences (mean gap between samples of 73.7 years over 8–0 ka) compared to MW (63.2 years), which could limit the detection of the 200-year cycle and allow the dominance of the 400–500 year cycle caused by oceanic forcing and/or arising as a subharmonic of the 200-year cycle.

## Causes of Holocene millennial drought in eastern North America: comparison with prior studies

In our water balance reconstructions, the patterns for PC1 and PC2 sample values across time (Fig. 3c, e) parallel those found earlier for a dataset of pollen and alkenone based summer temperature reconstructions, lake-sediment stable isotope records, and lake-level and dust-flux moisture proxy data, spanning the mid-latitude US from

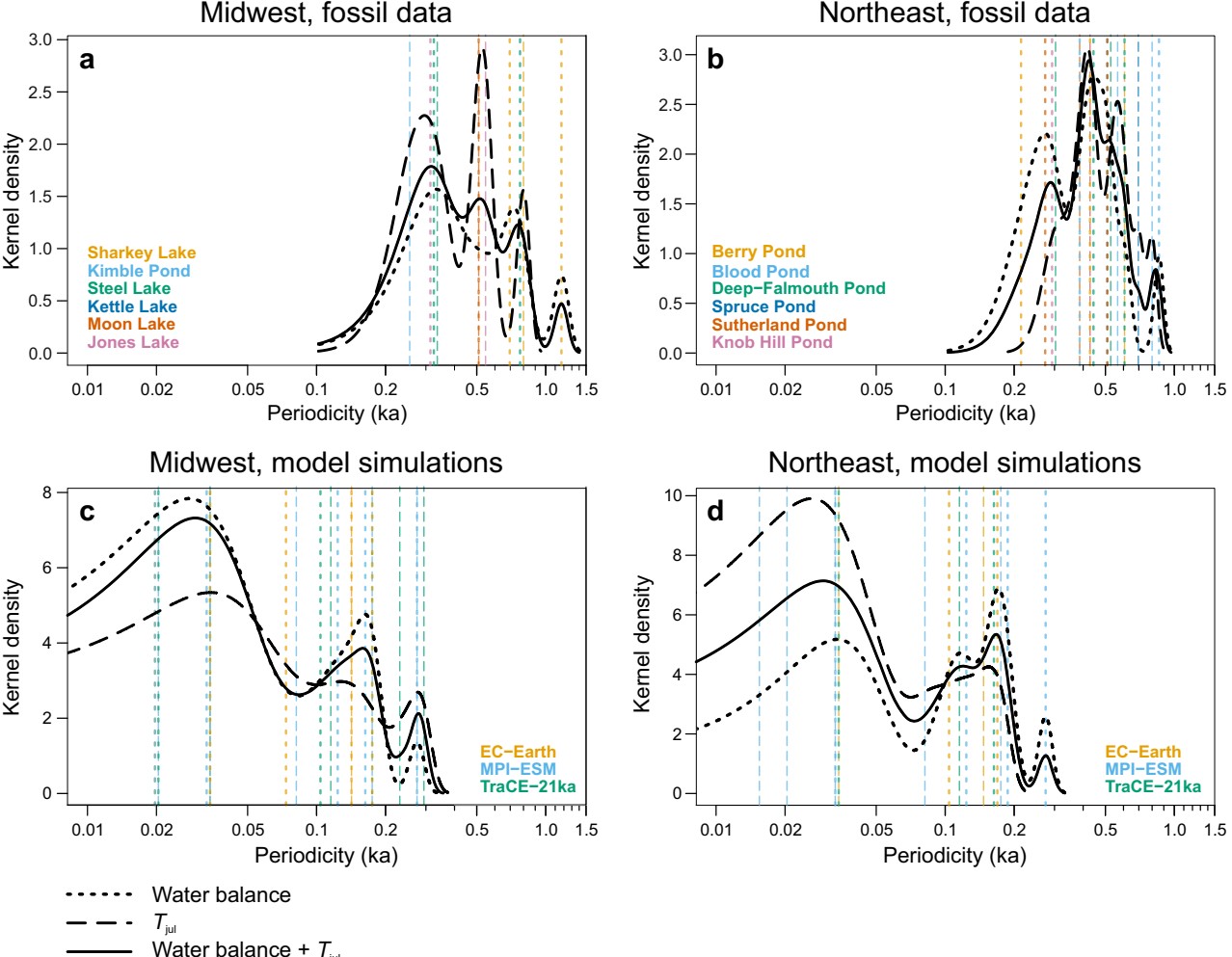

**Fig. 6 | Periodicities in proxy datasets and model simulations.** Distribution of significant periodicities between 8.0 and 0.25 ka is shown for six high-resolution fossil pollen sequences from the (**a**) Midwest and (**b**) Northeast regions, and for the three transient model simulations in the (**c**) Midwest and (**d**) Northeast. The *thin vertical lines* mark significant periodicities found for either water balance (*dotted line*) or July temperature (*dashed line*), with the color identifying the fossil site or climate model. The *thick black curves* show kernel density estimates fitted to the significant periodicities found for water balance (*dotted line*), July temperature (*dashed line*), or both combined (*solid line*). All detected periodicities are presented in Supplementary Table 3, the wavelet power spectra of the individual reconstructions and simulations in Supplementary Figs. 18–25, and the mean power plots in Supplementary Fig. 26.

the Rocky Mountains to the northeast Atlantic margin[12]. These PC1 and PC2 patterns closely correspond to two key forcings affecting this region: summer insolation (PC1; Fig. 5c) and the LIS deglaciation, which is closely associated with an increasing east-west moisture contrast (PC2; Fig. 5d)[12]. However, PC1 deviates from summer insolation before 8 ka due to the presence of a sizable LIS (Fig. 5c), while the PC2 trend reverses after 5 ka (Fig. 5d) due to a greater moisture increase in MW compared to NE, as the MW recovers from the later drought maximum in the mid-continent (Fig. 5a), which now starts to decrease the longitudinal moisture gradient captured by PC2.

Our reconstructed drought pattern, with the mid-continent drying over 10–8 ka while the NE gets wetter, also aligns with earlier reconstructions based on lake-level reconstructions, drought-indicating depositional hiatuses, and pollen abundances of drought-resistant and mesic taxa[15]. This spatiotemporal pattern has been hypothesized to be explained by LIS deglaciation, and a following waning of the glacial anticyclone and an increased influence of the Bermuda subtropical high, leading to a rerouting of the northward moisture advection towards the NE region and away from the mid-continent[1,8,15,52]. However, earlier climate simulations (represented here by TraCE-21ka; Fig. 2a) have shown inconsistent patterns of hydroclimatic change compared to proxy data[9,10], and notably, indicating

greater than modern precipitation and moisture balance (ratio of actual vs. potential evapotranspiration) at 6 ka through the U.S. mid-latitudes[53]. This poor data-model agreement has made it difficult to definitively attribute reconstructed hydroclimate variations to specific atmospheric dynamics.

Now, our climate reconstructions for the eastern North-American mid latitudes show good agreement with the latest generation model simulations regarding the first-order patterns of drought initiation and progress over the early and middle Holocene. Thus, we are now better positioned to discuss the changes in water balance and their drivers. As the data–model convergence resolves the fundamental mismatch seen in earlier studies, we suggest that the major spatiotemporal moisture patterns are adequately explained by the summer insolation trend combined with the Laurentide Ice Sheet deglaciation and its effect on atmospheric circulation.

### Role of the Laurentide Ice Sheet (12–9 ka)
Based on the CESM1 simulations for the early Holocene, the distinct wet conditions at 12 to 11 ka can be largely attributed to a substantial reduction in PET, by -30 to 40% ( -180 to 230 mm) at 12 ka relative to preindustrial levels. This reduction in PET exceeds the decline in $P$ during the same period, estimated at -15 to 25% ( -120 to 180 mm)

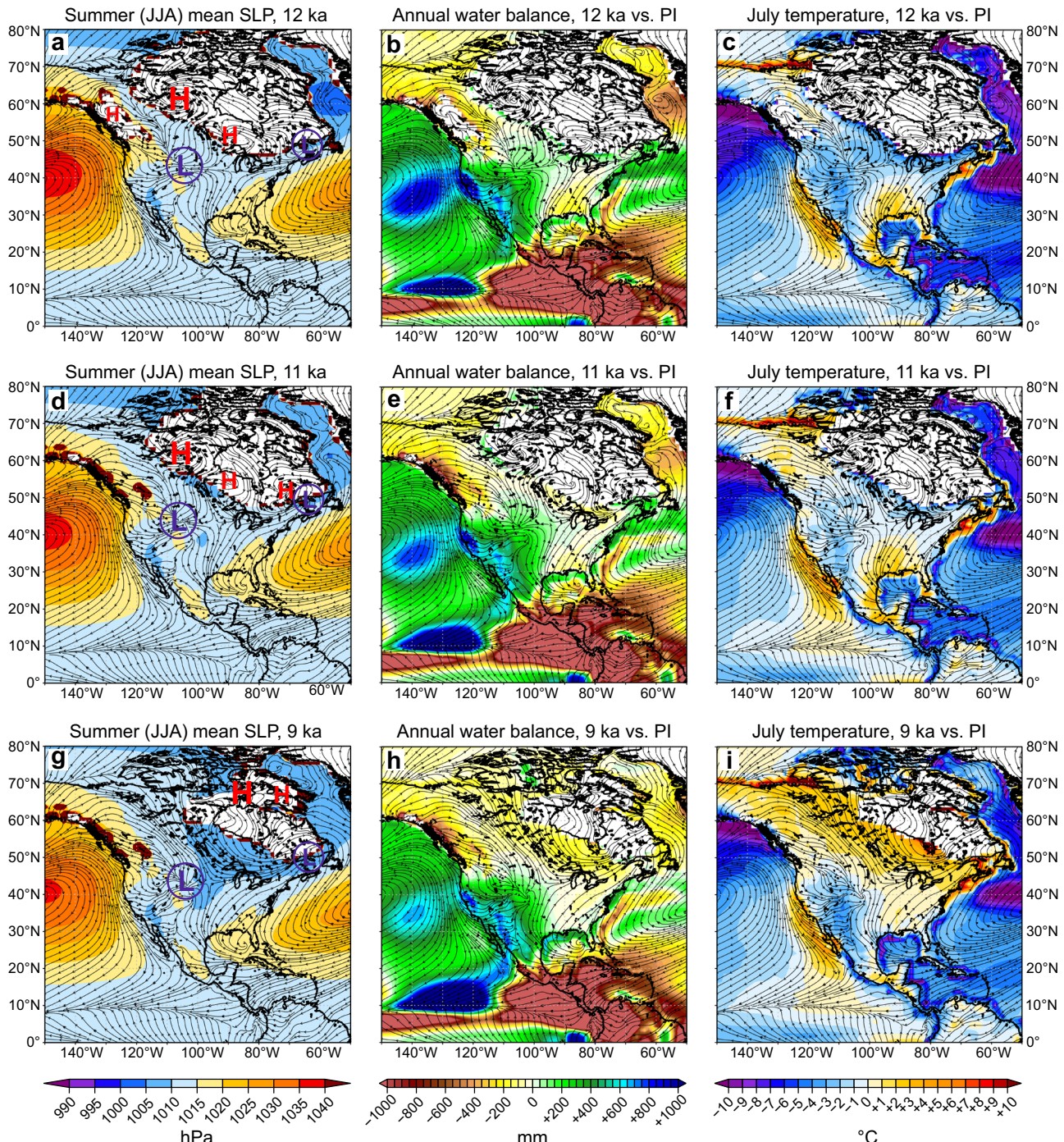

**Fig. 7 | Simulated North American climate anomalies of water balance and $T_{jul}$ relative to preindustrial and governing mean pressure states during the early Holocene, based on CESM1.** The panels show mean sea-level pressure (**a**, **d**, **g**), annual water balance anomalies vs. preindustrial (PI) (**b**, **e**, **h**), and July temperature anomalies vs. PI (**c**, **f**, **i**), for three time windows: 12 ka (**a**–**c**), 11 ka (**d**–**f**), and 9 ka (**g**–**i**). The *black arrows* indicate the summer wind fields and *white areas* the extent of prescribed North American ice sheets.

(Fig. 4a). The strongly negative ΔPET is consistent with dominating cold and dry winds from northwest to east due to anticyclonic blocking over the LIS at 12 ka and 11 ka (Fig. 7a, d), where strong summer cooling with low PET overcompensates the significantly reduced precipitation, resulting in positive water balance anomalies (Fig. 7b, e). This is consistent with several degrees colder $T_{jul}$ in both the reconstructions (Fig. 2b) and the CESM1 simulations (Fig. 7c, f).

The dipole of very wet conditions in MW vs. dry in NE in the reconstructed water balance around 12 and 11 ka BP (Fig. 5a) can be explained by dynamical changes in atmospheric circulation linked to

anticyclonic blocking over the ice sheet (Fig. 7a, d). To further explore the linkages between the regional water balance and large-scale atmospheric circulation, we performed a canonical correlation analysis (CCA) between large-scale variations of summer sea-level pressure (SLP) over North America and regional variations in water balance ($P - PET$) seen in the CESM1 runs for 12, 11 and 9 ka (Fig. 8). In general, CCA can identify pairs of linear combinations (canonical variates) from the two sets of variables that are maximally correlated with each other (see "Methods" for details). Applied to water balance in our study region, the CCA identifies strongly correlated patterns between

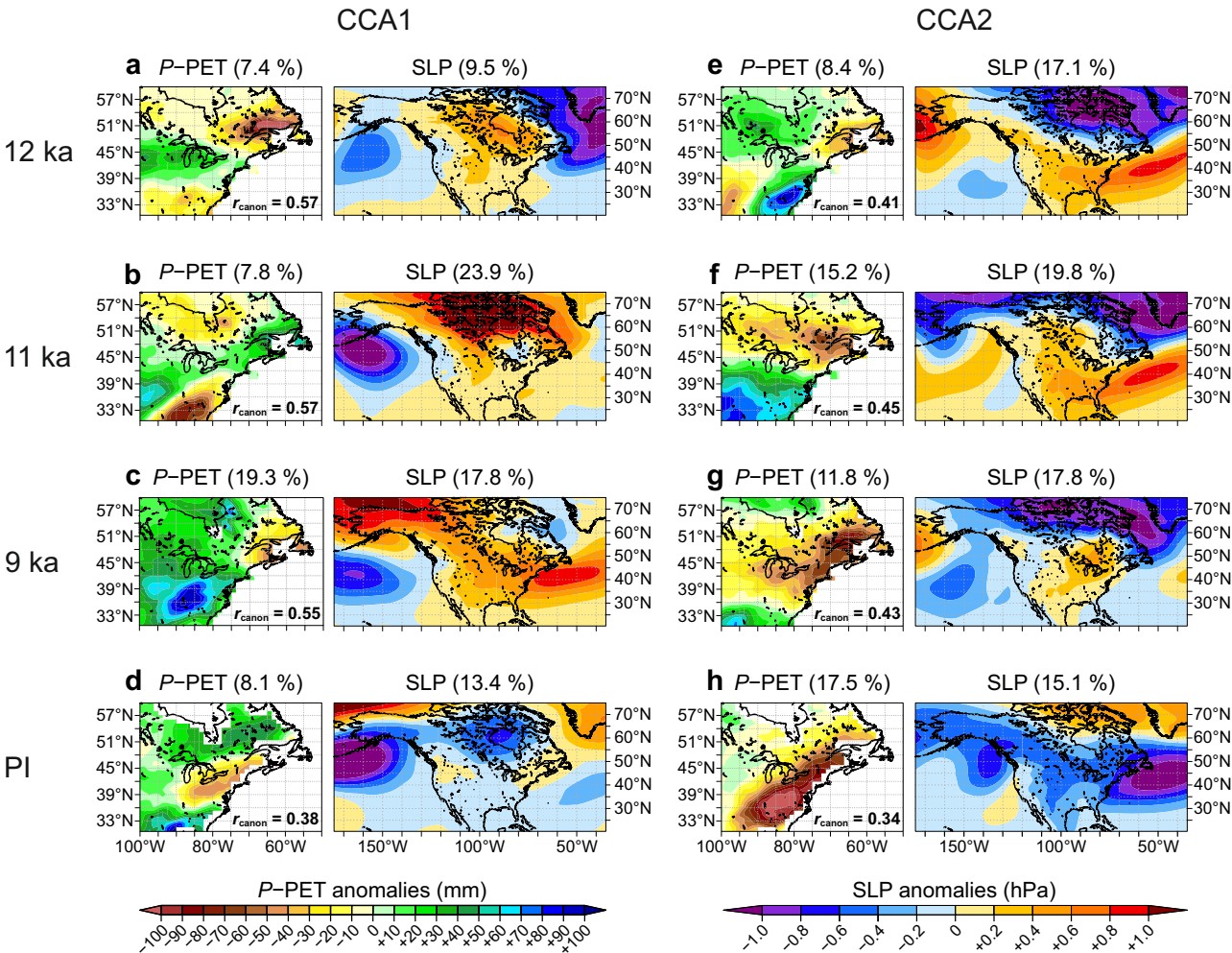

**Fig. 8 | Joint variability of water balance and sea-level pressure.** Changes in joint variability patterns in canonical correlation analysis (CCA) between simulated annual water balance (annual precipitation minus potential evapotranspiration; $P-PET$) and simulated summer (June, July, and August) sea-level pressure (SLP) are shown based on CESM1. Results are shown for 12, 11, and 9 ka, as well as the preindustrial (PI) control run. For each of the time windows, the results for the first CCA component (CCA1) are shown in panels (**a**–**d**) and for the second CCA component (CCA2) in panels (**e**–**h**). CCA1 shows the result with the highest, CCA2 with the second highest correlation ($r_{canon}$; indicated inside each $P-PET$ panel) between water balance and SLP. Note that the CCA1 may not always represent the pattern with highest explained variance (indicated in parentheses in the panel titles) for the individual $P-PET$ or SLP component, which in some cases is larger for CCA2, as the analysis was targeted to find the strongest modes of joint (rather than individual) variability in $P-PET$ and SLP. All $r_{canon}$ are statistically significant with $p < 0.05$.

variations in large-scale SLP and regional $P-PET$, suggesting that shifts in summer SLP are integral to the observed changes in water balance. As suggested by a decomposition of hydrological components of the water balance in model simulations (Fig. 4), the annual water balance variations can be expected to be linked to variations in summer and hence summer SLP, which is consistent with a comparable CCA for Europe[54].

In our CESM1 simulation, the reconstructed water balance dipole around 12 ka BP is consistent with the pattern identified by CCA1 (7.4% explained variance) with SLP. The contrast between the wet MW and the dry coastal NE is caused by a meridional SLP anomaly pattern identified by CCA1 (9.5%) with positive anomalies over the ice sheet representing the anticyclonic blocking (Fig. 8a). Even CCA2 (8.4%) for the water balance yields such a dipole pattern, which is now linked to a meridional pressure pattern of SLP in CCA2 (17.1%) (Fig. 8e) that effectively steers humid air into the MW while dry air deflects precipitation in the NE (Fig. 7b). The west–east water balance dipole weakens at 11 ka, as the blocking anticyclone retreats northward with the retreating ice sheet, allowing more humidity to reach the Northeast (Fig. 8b). Along with the anticyclonic pattern, the meridional SLP

pattern also moves northward but keeps humidity low to the south and southwest of the ice sheet (Fig. 8f).

By 9 ka BP, the anticyclonic variability pattern is replaced by more zonal SLP patterns due to the almost vanished ice sheet (Fig. 8c). The persisting dry conditions in NE seen in reconstructions (Fig. 5a) appear to be caused by the northward-shifted meridional SLP variability (indicated by CCA2 of SLP in Fig. 8g) compared to the modern climate (Fig. 8h). Note that even at 9 ka the co-variability pattern between SLP and water balance is very different compared to preindustrial, where the water balance is dominated by the cyclonic activity over north-central Canada (Fig. 8d). With the last remaining parts of LIS retreating to the far north at 9 ka, the climate over northeastern North America rapidly switches from the blocked deglaciation pattern seen at 12 and 11 ka (Fig. 7a, d) towards a regime driven by summer orbital forcing. This period marked the onset of increasing aridity, as summer warming (Fig. 7i) leads to an increase in PET that now exceeds $P$ (Fig. 7h), intensified by the shift from cool maritime winds to warm continental westerlies due to the LIS retreat that removes the blocking anticyclonic flow from the region (Fig. 7g). Overall, these CCA results based on CESM1 confirm the hypothesis[1,8,15,52] that the dipole in water balance in

the early Holocene is indeed driven by anticyclonic blocking over the ice sheet.

### Insolation-driven drought (8 ka onward)

The disappearance of the LIS by 8–6 ka (Fig. 5d) marked an end of the influence of the ice sheet on the spatiotemporal hydroclimate patterns, and from the mid-Holocene drought and its easing towards the present were largely driven by long-term changes in orbital summer insolation (Fig. 5c). Warming and drying rapidly increased to maximum drought conditions at 8 ka BP where the strong warming caused higher PET than today, while annual precipitation, despite a gradual rising trend, remained almost as low as during the late deglaciation (Fig. 4). This implies that drought is driven by both the higher PET, due to much warmer summer temperatures in response to orbital forcing, and clearly lower annual $P$ than under preindustrial conditions.

The (relative) drought conditions through much of the Holocene, as delineated by both our reconstructions and the MPI-ESM and EC-Earth simulations, suggest that the recent and modern climate is unusually wet while drier conditions seem to be the norm during most of the Holocene. This period of enhanced aridity was largely due to elevated summer insolation, temperature, and evaporative demand. Future simulations project that, as greenhouse gas concentrations rise and global temperatures increase, midcontinental North America will experience an increase in precipitation yet a decrease in plant-available soil moisture, because of enhanced evaporative demand[5]. It is hence plausible that the reversal of the natural neoglacial cooling observed over recent millennia through anthropogenic global warming might cause a return of midcontinental aridity in eastern North America, for which the early to mid Holocene serves as a reasonable natural analog.

## Methods

### Pollen data

For pollen–climate calibration data, we use an eastern North American dataset[37] derived from the North American Modern Pollen Database[55,56], with the addition of 165 modern samples from the Neotoma Paleoecology Database[57] (https://neotomadb.org/) and originating from more recent work[22,58,59], to improve the coverage in the western prairie. The dataset (Fig. 1; Supplementary Data) includes 2419 surface pollen samples with climate means extracted from CRU CL v. 2.0 climate grids for 1961–1990[60]. Correlation between water balance and $T_{jul}$ is low ($r = -0.11$; Supplementary Fig. 14) due to the near-orthogonal gradient directions (north-south for $T_{jul}$, east-west for water balance), facilitating the independent modeling of temperature and moisture signals in the modern pollen data. The pollen taxonomy follows the 64-type list of ref. 38 for eastern North America, except for *Pinus* (pine) which is combined in a single column because the fossil pollen data used do not consistently differentiate between diploxylon and haploxylon pine. The calibration dataset excludes the region dominated by the ecologically distinct southeast pine species[18,20]. Importantly, this taxonomy does not include *Ambrosia* (ragweed), which is an important Holocene prairie taxon and indicator of hydroclimate variability[26]. However, *Ambrosia* has a strong human impact on its present distribution[17] potentially biasing pollen–climate calibration models or leading to challenges in interpretation of paleoclimate reconstructions[22].

To study the spatio-temporal patterns in drought, while pooling sufficient data to establish robust paleoclimate signals, we assembled fossil pollen sequences for the three regions (Fig. 1) from Neotoma. The following criteria were used to filter for acceptable fossil pollen sequences: at least 30 pollen samples and five absolute datings, to enable the detection and alignment of sub-millennial climate signals, a modern water balance within the range of the calibration data (−348 to 1343 mm) by a margin of at least 150 mm (Supplementary Fig. 14), to ensure that past variation relative to modern conditions can be reconstructed, and bottom age of at least 10 ka. The number of sites

meeting these criteria was 20 for MW, 18 for GL, and 28 for NE (Supplementary Table 1). The MW and NE clusters each contained six sequences with considerably higher than average number of pollen samples (minimum 125) and [14]C dates (minimum nine, excluding dates rejected by the authors), and we prepared separate reconstructions from these sets of well-dated high-resolution sequences. In MW, all six high-resolution sites use AMS [14]C dates from plant macrofossils or charcoal, while the NE high-resolution sites largely use AMS dates from bulk sediment.

### Paleoclimate reconstruction

The BRT models for pollen–water balance, pollen–$T_{jul}$, and pollen–GDD5 calibration were created using the R[61] library *gbm*[62] with the following settings: maximum number of trees = 3000, learning rate = 0.025, tree complexity = 4, bagging fraction = 0.5. For further details on these models, see ref. 37.

To align reconstructions from different modern climatic settings, all reconstructions were expressed as deviations from the site-specific mean over the past 4 ka. To equalize the impact of fossil sites on the reconstructed moisture and temperature levels regardless of their sampling resolution, we then interpolated all site-specific reconstructions at 50-year time step. Synthesis reconstructions were then calculated for each region as the mean of all reconstructions, repeated 1000 times using bootstrap samples of all fossil sequences to calculate the 95% errors (2.5th and 97.5th percentiles) of the mean. SiZer maps[63] (implemented using the R library *SiZer*[64]) were calculated based on the synthesis curves to identify significant rising and falling trends in the reconstructions when smoothed at a range of decadal-to-millennial bandwidths. To further analyze spatiotemporal patterns in the water balance reconstructions, we performed a principal components analysis (PCA) on the reconstructions from all sites covering the 11–0.25 ka time span (53 sites out of 66)[12].

In our reconstructions for the MW and NE regions, we primarily use the reconstructions based on the high-resolution subsets of sites. While the reconstructions using all sites are broadly similar and would not change the key interpretations regarding the patterns and drivers of multi-millennial moisture change, the all-site reconstructions appear to truncate the Holocene range of water balance variation, while the high-resolution reconstructions also reveal deeper sub-millennial moisture and temperature variations between the sites (Supplementary Fig. 15). However, the all-site reconstructions for MW and NE may be preferable for future studies concerned with multi-millennial moisture or temperature levels, which are likely to be sufficiently captured by all sites which passed our initial filtering, and we have thus also included the full results using all sites in the Supplementary Data.

To explore the effect of calibration model selection, we also prepared the water balance and $T_{jul}$ reconstructions using MAT[21], the most commonly used approach in pollen-based climate reconstructions in North America[12,19,38]. The MAT was implemented using the R library *rioja*[65] and a weighted mean of five best analogs. The timing of the early-mid-Holocene multi-millennial droughts and the temperature maximum is broadly consistent between reconstructions prepared with BRT and MAT (Supplementary Fig. 16), showing that the qualitative patterns underpinning our main conclusions are robust to calibration method selection. However, the BRT reconstructions show a stronger peak warming in all sectors, while the broadscale Holocene drought is deeper in MW and GL but weaker in NE compared to MAT-based reconstructions. We primarily employ the BRT-based paleoclimate reconstructions in this work, because in cross-validations using a robust $h$-block scheme[37], the BRT model performs considerably better for water balance (coefficient of determination 0.65 [BRT] vs. 0.56 [MAT]) with a smaller advantage in $T_{jul}$ (0.88 [BRT] vs. 0.86 [MAT]) (Supplementary Fig. 17). Further, earlier $h$-block cross-validation experiments using a wide range of $h$ values show that the

performance advantage of BRT over MAT widens with large $h$, suggesting more robust performance of BRT with fossil samples with poor modern analogs.[37]

The BRT models gain additional support from an analysis of the relative contributions of the most important predictor taxa (Supplementary Figs. 12,13) in the pollen−climate calibration data (Fig. 1). This analysis confirms the ecological realism of the BRT-based calibration models, with the water balance model relying heavily on the prairie forbs *Artemisia* (sagebrush) and Amaranthaceae (amaranth family). The five most important water balance predictors also include moisture-sensitive trees, including the comparatively drought-resistant *Pinus* (pine) but also the moisture-demanding *Betula* (birch) and *Abies* (fir)[18,20], which the BRT models employ in this order as indicators of progressively wetter conditions (Supplementary Fig. 12). By comparison, the $T_{jul}$ model uses largely different predictors (Supplementary Fig. 13) with a predominant driver in eastern *Quercus* (oak), a well-understood summer temperature-sensitive tree in eastern North America[17,18,20,56]. The water balance and $T_{jul}$ calibration models thus have distinct structures (i.e., rely on different paleobotanical signals) and are consistent with prior ecological knowledge on indicator species for summer temperature and moisture availability.

## Paleoclimate model simulations

We compare our reconstructions for water balance, $T_{jul}$, and GDD5 with a set of climate model simulations. The model outputs were extracted using boxes coinciding with the pollen site clusters (MW: 42−51°N, 92−100°W; GL: 40−47°N, 85−90°W; NE: 40−47°N, 69−75°W, excluding ocean), shown in Fig. 1. As a reference simulation covering the whole period, we reconsider the (quasi-)transient, fully coupled atmosphere-ocean simulation TraCE-21k that covers the full period from 22 ka to the modern climate[66,67]. The simulation incorporates changes in orbital and greenhouse gas forcing, adjustments of continental ice sheets, and sea-level rise, as well as meltwater fluxes triggering abrupt climate shifts in agreement with geological records. The CCSM3 model used for TraCE-21k has a relatively coarse spatial resolution of 3.75° × 3.75° ( ~400 km) and was found to produce an inconsistent evolution of North American water balance in a previous study for the Holocene[9,10].

To evaluate whether a higher spatial resolution alters the representation of atmospheric flow and related climate in response to the LIS during the period 12 to 9 ka, we analyze additional 100-year snapshot simulations conducted with CESM1.0.5 with a ~4× higher horizontal resolution of 0.9° × 1.25° ( ~100 km) for the periods 12 ka, 11 ka, and 9 ka. The model setup and boundary conditions are adjusted to these periods for changes in radiative forcing, ice-sheet configurations, and sea-level change (described in more detail for 13 and 12 ka in ref. 69). The ocean and sea-ice states for these periods are prescribed from TraCE-21k using a mean annual cycle climatology calculated as ± 50 years around the target period. The high spatial resolution leads to fundamental differences in response to the presence of ice sheets over Europe[68,69] compared to its parent model CCSM3 despite having identical radiative forcing and ocean states. Simulations with CESM1 for the deglaciation show a higher sensitivity for changes in the hydrological cycle and tropical convection with more extreme changes in seasonality compared to CCSM3[70,71].

For the remaining transient evolution of the Holocene, we make use of simulations covering the period 8 ka BP to modern conducted with the MPI-ESM 1.2 model[40,41] and EC-Earth3-veg-LR[72,73]. The MPI-ESM (Earth System Model of the Max Planck Institute for Meteorology) is forced by variations in orbital forcing, greenhouse gases and solar irradiance[40], with an updated volcanic forcing as well as land-use and dynamical vegetation changes[41] with a horizontal resolution of 1.875° × 1.875° (nominal resolution of 1.5° for ocean and sea-ice). EC-Earth is forced by orbital and greenhouse gas changes at a horizontal model resolution of 1.125° × 1.125° (1° for ocean and sea-ice).

For comparison with our reconstructions, we use simulated annual water balance, $T_{jul}$, and GDD5. Annual water balance is calculated from simulations as $P − PET$ [mm/a] where $P$ = annual total precipitation [mm/a] and PET = potential evapotranspiration. To be consistent with the calculation used in the pollen−climate calibration data[37], we calculate PET as $58.93 × MABT$ [mm/a] where MABT is the mean annual biological temperature derived as the annual average of monthly means > 0 °C. Similarly, GDD5 was calculated from model outputs using the same formula as for the pollen−climate calibration data, with mean daily temperatures interpolated from a sine regression fitted to the monthly mean temperature values[37]. Anomalies of water balance, $T_{jul}$, and GDD5 are calculated relative to the preindustrial climate (1850 AD) for CESM1 and relative to 0.25−0.75 ka (as done for reconstructions) for transient simulations of CCSM3, MPI-ESM and EC-Earth. Additional model output from CESM1 is used for dynamical analysis for sea-level pressure [hPa] and near-surface wind at the terrain following lowest sigma-hybrid coordinate level 992 ( ~20−50 m above ground). The pressure data for past climate states were corrected for systematic offsets by subtracting the difference of the global mean sea-level pressure between climate states in the past and preindustrial to avoid spurious offsets caused by non-dynamical changes of lower sea-level and glacio-isostatic land movements and ice sheets.

To investigate whether the presence of the retreating LIS in the early Holocene is indeed a controlling factor for regional changes in water balance[12,15], we perform a Canonical Correlation Analysis[74] (CCA) of simulated water balance $P − PET$ variations with the large-scale atmospheric circulation over North America for the periods 12, 11 and 9 ka BP (Fig. 8). As annual precipitation $P$ changes roughly linearly across the 12,000 years and evapotranspiration PET is ~0 during months ≤ 0 °C and hence dominated by summer changes (Fig. 4), the CCA is performed on regional annual $P − PET$ vs. large-scale summer SLP using monthly data from CESM1.

The CCA results in two patterns showing maximum correlation among two geophysical fields X and Y. In our example X relates to SLP and Y relates to $P − PET$. Both variables are derived from CESM1 simulations for different periods of time, 12, 11 and 9 ka BP. The hypothesis is that basic (tele-)connection and structure between the different periods changed because of (profound) changes in background surface boundary conditions regarding continental eastern Laurentide ice sheets. These changes should then also be reflected in the canonical patterns between SLP ($X_{CCA}$) and $P − PET$ ($Y_{CCA}$).

A technical pre-processing of the original fields X and Y relates to an empirical orthogonal function (EOF)-truncation prior to carrying out the CCA. This simplifies the mathematical structures. In our case, five EOFs representing ~60% of total variance in $P − PET$ and ~72% in SLP are used as input to generate the CCA patterns. One of the features of the CCA patterns is that their time coefficients are orthogonal, i.e. they do not show any temporal correlation.

Concerning the interpretation of the CCA results, the leading CCA1 patterns share the highest correlation among the variables X and Y, whereas the higher indexed CCA patterns show lower correlations. Therefore, higher indexed CCA patterns should be interpreted with care, especially when their canonical correlation indicates low values. Here, we focus on the first two leading patterns.

## Analysis of periodicities

To assess the presence of significant periodicities, wavelet analyzes (Morlet wavelet) were performed for both paleoclimate reconstructions and the transient model simulations using the R library *WaveletComp*[75]. The wavelet analysis was performed individually on each of the water balance and $T_{jul}$ reconstructions prepared from the subsets of high-resolution sites available for the MW and NE site clusters (12 sites total; Supplementary Table 1) using the BRT calibration models. For comparison, we also performed the wavelet analysis

on the simulated water balance and $T_{jul}$ anomalies from each transient model simulation ($N = 3$) for the spatial domains of the MW and NE site clusters. In all wavelet analyzes, we considered the 8–0 ka time span covered by all reconstructions and transient simulations. Prior to the wavelet analysis, the transient model simulations data were averaged using a 50-year running mean to mimic the 50-year resolution of the paleoclimate reconstructions, using the R library *zoo*[76]. To remove oscillations on the millennial timescale, all data (paleoclimate reconstructions and the transient model simulations) were filtered using a high-pass red noise Butterworth filter using the R library *signal*[77] (cutoff = 1000 years). Finally, the significant periodicities ($p < 0.05$) relative to red noise observed in the wavelet analysis were summarized in a kernel density plot.

## Data availability

The paleoclimate reconstructions, the model simulation outputs, and the pollen–climate calibration dataset generated in this study have been deposited in the Figshare database under accession code 28482608.

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

## Acknowledgements

This study was supported by Research Council of Finland projects 331426 (J.S.S.) and 334509 (A.L.L.D), Swedish Research Council for

Sustainable Development (FORMAS) projects 2020-01000 (F.S.) and 2023-01631 (F.S.), Swedish Research Council (Vetenskapsrådet) projects 2022-03129 (Q.Z.) and 2017-04232 (Q.Z.), and the U.S. National Science Foundation project DEB-1856047 (B.S.). The simulations with EC-Earth3-LR, CESM1 and data analysis were performed using Swedish National Infrastructure for Computing (SNIC) and National Academic Infrastructure for Supercomputing in Sweden (NAISS) at the National Supercomputer Center (NSC), partially funded by the Swedish Research Council institutional grants 2018-05973 and 2022-06725 to Bolin Center for Climate Research. Open access was funded by the Helsinki University Library.

## Author contributions

J.S.S., F.S and J.W.W. had the main responsibility in designing the study and writing the manuscript. J.S.S., M.L., J.W.W., and B.S. conducted the data synthesis. J.S.S. prepared the climate reconstructions. F.S., Q.Z., and J.J. performed and post-processed the Earth system model simulations, and F.S. and S.W. analyzed their outputs. A.L.L.D. prepared the wavelet analyzes. B.S., J.W.W., and J.S.S. reviewed the existing research on the Holocene hydroclimate of North America. J.S.S., F.S., J.W.W., B.S., A.L.L.D, S.W., J.J., Q.Z., and M.L. participated in analyzing the results and contributed to manuscript development.

## Competing interests

The authors declare no competing interests.
