## [Transparent Peer Review file · Nature Communications]

Patterns and drivers of Holocene moisture variability in mid-latitude eastern North America

Corresponding Author: Dr Sakari Salonen

Version 0:

Reviewer comments:

Reviewer #1

(Remarks to the Author)
Dear Editor,

I would like to first apologize to the authors for delays in my submission of the current review. I have read the manuscript from Salonen and al. with great interest and found it interesting and worthy of publication in your journal after revision. While I have reservations on some methodological aspects described below that have to be addressed, I do not think it would fundamentally alter the conclusions of the paper, which rely on several lines of evidence. I have thus a few general comments on the content and structure of the manuscript, and a list of detailed comments with line numbers.

Structure wise, I think that the manuscript could be generally condensed a little both in terms of text and figures. I also note that it seems to me that new results are integrated in the discussion. I don't think the classic results-discussion separation is necessary and that the sections could be restructured with more descriptive titles instead.

My general comments have to do mainly with (i) the handling of the reconstructions, specifically relating to the stacking and resolution, and (ii) with the analysis of periodicities.

(i) If I understand well the stacking methodology, running 250-year windows at 50-year intervals were used to average all the available data points. This means that a high-resolution series with say 5 points in the window would contribute 5 times more than that of a low-resolution series with only 1 point in the window, right? Not to mention that some series might not even contribute to some windows if some of their points are more than 250-year apart. This can lead to apparent non-stationarity in the variability as more estimates are available in the more recent period and thus, on one hand more noise is averaged out, and on the other hand, more smoothing occurs due to time-uncertainty. In both cases, we thus expect less variability (and not more) for stacks of higher resolution series, unless the signal-to-noise ratio is high and the time-uncertainty is low. I would thus suggest to at least perform a temporal interpolation before averaging the series as was done for example by <https://doi.org/10.1038/nature25464>.

Is there evidence that BRT improves the signal-to-noise ratio (SNR) within clusters compared to MAT? It seems clear that the variability of the BRT-based stacks is higher than the MAT-based ones, but is this due to higher SNR with BRT, or simply because the variance of BRT reconstructions are generally higher than that of MAT ones? In other words, is the variance ratio from individual series in clusters to their stack the same for both methods, or is it really higher (lower) for MAT (BRT)?

(ii) While the analysis of periodicities seems to be computationally correct, I have reserves on the methodological choices. The methods used have however been used quite frequently in palæoclimate studies and therefore my criticism is not necessarily specific to the author's work. First, I have reserves on the usage of wavelets to obtain time-dependant and timescale-dependent variability estimates from palæoclimate data, the issue being that we are extracting a lot of information from relatively short and noisy timeseries, and changes in periodicity through time is likely random rather than expressing a real change in the state. For such Holocene timeseries, resolving about one order of magnitude in frequencies, more robust results would be obtained by looking at the wavelet (or regular power) spectrum for the whole time period (e.g. Fig.1 in <https://doi.org/10.1038/s41561-022-01056-4>).

Wavelet spectrogram can still be qualitatively informative in this case to see how the variability is distributed across time and frequency, but to infer quantitative information requires some additional considerations. The null hypothesis has to describe the "background variability" correctly. While an AR1 red noise is commonly implemented and used in palæoclimate analysis, it almost never describe the background variability satisfactorily and leads to spurious estimation of peak significance (e.g. <https://doi.org/10.1175/JCLI-D-22-0011.1>). Therefore, most, if not all, of the periodicities are likely not significant (and with $p < 0.05$ we still expect 5% of false positives). I also note here that I wonder how the periodicities were identified from the spectrogram as they are not stationary time, was that from a mean wavelet spectrum?

I also have reserves with respect to the frequencies that could be resolved given the low- and high-pass filters that were applied. The pollen data is computed with 250-year sliding windows, which corresponds to a convolution with a rectangular window, or with the multiplication of a sinc function in the Fourier domain. What this means really, is that the cutoff frequency affected by the sliding window, effectively a not so ideal lowpass filter, is greater than $1/(500 \text{ years})$. Then on top of that, a high-pass filter is used to kill the power below $1/(1000 \text{ years})$, and thus we have a very narrow window between $1/500$ and $1/1000$ years where variability estimates are less affected by the smoothings (albeit the running mean filter has a sizeable impact even on the $(1/1000 \text{ years})$ frequency). What is then the sense of the AR1 null hypothesis when the shape of the spectrum does not represent the signal but the smoothing filters right? In the end it looks like the identified periodicities are quite randomly distributed over the range of frequencies and I'd suspect that any similarity, albeit I fail to see the described broad agreement, would be more likely an interaction between the shape of the filters and that of the null hypothesis.

I think that in order to evaluate whether the water balance and July Temperature are linked, I would suggest a cross-spectral analysis. If common periodicities do exist, they should be clearly evidenced by such analysis as significant levels of coherency should be attain at the relevant frequencies.

Below are detailed comments with line and figure references:

Line 72-76: A bit of a weird sentence, you say MAT and responses surfaces were used before, and that not MAT and other classical methods are used. I'm not sure what the logic is, except to say that response surfaces might not be used so much anymore? I just think the sentence could be simplified to say "different methods such as X and Y have been developed and used."

Line 78: Last two sentences could be combined and shortened, and a summary sentence could be added to the paragraph with the key takeaway point.

Line 82: Is it really abrupt if it occurs over a millennium? Abrupt kinda implies a departure from more "normal" variability to something more extreme, I'm not sure if that's really the case, at least not when we look at the average of several reconstructions (which might be overly smoothed) but on the other hand single reconstructions are likely containing lots of non-climatic noise.

Line 94: The time is always ripe for a new and better analysis. I think a "Why" could be included here too already, "because of new methods and data".

Line 95-96: "have arisen" sounds like they just appeared on their own, they were developed (by people).

Line 118: transient climate model simulations? I would maybe use the phasing: "an ensemble of N Holocene transient simulations across 4 climate models"

Line 125: No need to repeat there are four sets of model simulations here, just "the Holocene simulations"

Line 128: Finally instead Fourth?

Line 129: Maybe skip the "concerning the spatiotemporal patterns.... America" and just say directly "we test the hypotheses that the patterns are linked to..."

Figure 1: Just wondering what the reason is for the lack of records between the Great Lakes cluster and the Northeast cluster? Lack of high resolution records for these areas or because the signal is too mixed there and not useful for the analysis? I would also suggest to move the map to the extended data figures.

Line 150: Typo "the MD" maybe "in the MD"?

Line 151: Typo "at a 10-7 ka"

Line 159: Unclear what means that the progress of the drought is well tracked by proxy data. Do the authors simply mean that negative water balance anomalies are reconstructed for all regions from 10ka onward?

Figure 2: Wondering if there are ways to simplify the figure and caption. Could we show the average of MPI-ESM and EC-Earth together and maybe only show MAT in the extended data fig? Please indicate the bandwidth h with units and without the logarithm on the axis (the spacing can remain logarithmic, but the labels would be clearer if given directly in ka). On the other hand, are those SiZer maps really necessary? What do they tell us except what we can already see on the plot? The GDD5 is maybe not necessary to be shown? Was there an additional CESM1 pre-industrial snapshot that was considered to

calculate the anomalies? Could the models be combined as on Fig 4, and maybe keep TraCE-21ka for the supplement?

Figure 2: Presumably the temperature and water balance ratio are relatively well correlated in the model simulations. Therefore, if water balance agrees with the reconstructions but not July temperature, I wonder whether a model that would simulate correctly the July temperature would then end up overestimating the water balance?

Line 190: Can the details of the comparison between BRT and MAT be moved to the methods or supplement to improve the readability flow?

Line 196: Maybe specify that the PCA is done across all the clusters.

Line 198: Maybe write "with the remaining components not exceeding 6.1%" rather than using a symbol " $< 6.1\%$ ".

Line 202-207: Could the description of PC2 be better and more simply explained? I just wonder a bit, what is the longitudinal water balance contrast, or the period of maximum contrasts due to peak multi-millennial drought. I think this could be said more simply. I don't know if the PC loadings can be said to be moving with longitudes, I see mostly two clusters with positive loadings in the interior and negative on the coast, and not so much transition in between.

Line 209: Are the two loadings covarying with the midwest sites doing so significantly? Given that the original series might have different resolutions (and I imagine the PCA is performed after interpolation), high amplitude loadings could still be random or dominated by noise.

Figure 3: What does "scores for each sample of water balance reconstructions" mean? Revise and clarify sentence.

Line 272: The shift from negative to positive PET anomalies looks like model bias, it's doubtful that a continuous curve could go through. Are there no simulations of CESM1 that overlap with the other runs to evaluate this? Again I wonder about how the CESM1 anomalies were computed and whether this could be related.

Figure 4: It's interesting to note that the CESM1 simulations show a wetter midwest while the transient ones show a wetter northeast. I wonder what we learn from the difference regional curves here other than this reversal between the models which doesn't seem to be interpreted. If we cannot trust the regional differences from the simulations, I would show instead one curve for the average across all three regions. Again, it would be interesting if an additional CESM1 simulation at 8ka and thus overlapping with the transient simulations was performed in order to see if the discrepancy is a model bias. I think the figure could be improved in general better evidence the key takeaway.

Figure 5: Revise figure. Either show the average spectra or the coherency to show common periodicities.

Line 344: Yet to prove with a coherency analysis.

Line 353: Missing closing bracket.

Line 362: Again unsure about the periodicity argument.

Figure 6: It seems that all the data shown here has been shown before on different panels, it would be nice if somehow we could condense the storyline and avoid duplication. (a) Seems like the Great Lakes and Northeast might have some anti-correlation, might show up on coherency analysis. (b) Why the shading under the curve? What high-pass filtering was done? Looks like the cutoff frequency might be too high, I would favour only detrending the very slow trend (could be done by subtracting a loess smoothing with a long span (on the order of a couple thousands of years)). (c) Seems that the deglaciation could explain the first part of both curves, and that the second part would be the insolation trend.

Line 387: How were the climatic impact of vertical land movements estimated? Why 0.5 K? Unclear to me how vertical land movements affect the climate.

Line 388: Why are they expected to negatively bias the HTM amplitude and delay reconstructed HTM?

Line 397: What is the contradiction?

Line 399: So are you suggesting that the July temperature reconstructions are of poor quality? Then why use them at all?

Line 404: What is meant by "significant" variations? Significant should be used in the context of testing.

Line 405: Are they however coherent between regions? I imagine that there is a high level of noise on single reconstructions, but still wondering on the robustness of centennial scale variations.

Line 412: Could the reason that the 5.5-5.0 ka wetting is visible across the three regions simply that it has higher amplitude and thus creates a stronger signal above the noise in all three regions, whereas elsewhere the centennial variability is masked by proxy noise?

Line 424: Were the series first interpolated before the average? The higher number of samples in the recent part means that

more noise will be averaged out and therefore, the difference in variability might be an indication that more proxy noise remains in the earlier part. In this sense, the higher data resolution should average more noise and decrease variability.

Line 429: 200-year is below the effective resolution of the series, which were produced with 250-year sliding windows. It does not make sense to interpret that timescale.

Line 430: Or maybe the significant periodicities around the 200-year frequencies correspond to a transition timescale of the AR1 null hypothesis where a mismatch is more likely to occur and lead to spuriously significant periodicities? That would be my guess.

Line 474: Period missing.

Line 493: What is meant by comparing large scale SLP and regional water balance? Do I understand right that in one case the CCA is performed over a region from longitude 170W to 40W and Latitude 20N to 75N and the other 100W to 40W and 32N to 58N only over land? These seem to be very arbitrary choices to me. Of course this type of analysis can be sensitive to the data that is included and choosing different domains can change the dipole spatial patterns observed.

Figure 8: Are we really learning something from all those figures? Could it be simplified? I am not so familiar with the details of CCA, but I wonder if the correlations are significant? I think the Methods on CCA could better describe the procedure.

Line 539: I struggle with this statement given that the rise in PET from 9ka to 8ka in model simulations was inferred from a change from CESM1 to transient simulations.

Line 616: Using 250-year windows means that the centennial scale variations, which were analyzed and interpreted, will be very smoothed. I think any variations below that should not be interpreted, at least in terms of their amplitude as they will be damped by construction. I think it would be better to interpolate to 50-year resolution all the series and stack them. The series could then be smoothed when plotting timeseries to evidence slower variability, but unsmoothed series should be used for wavelet/spectral analysis.

Methods

Line 572: comma before "which".

Line 573: Split sentence before "however".

Line 584: How were the cluster limits chosen? I wonder whether the gap between the GL and NE areas are a choice or a lack of records.

Line 587: Well-dated is good, but what does that mean concretely in terms of time uncertainty? Generally, I'd still expect a time uncertainty on the order of hundreds of years, meaning that stacking nearby sites would lead to a smoothing of the signal due to 'misalignment'. The effect can be seen on surrogate data created from model simulations with realistic time uncertainty and the resolution of pollen records on Ext. Data Fig 1 in <https://doi.org/10.1038/s41561-022-01056-4>. Of course the effect should be smaller in this case given the selection for high quality records, but I still suspect it to be important for the centennial variability. I also wonder here the impact of sediment mixing on the effective resolution of the high-resolution records

Line 616: Were the reconstructions interpolated before taking the rolling mean, as was done for example by Marsicek et al 2018, albeit here there would be no spatial gridding since we are only interested in stacking the clusters.

Line 625: Well it wouldn't be surprising that the stack of all records and high-resolution records show similar features as they are both using the high-resolution data, and if there is not interpolation before stacking done, then the high-resolution one will contribute more samples to the rolling mean and have a higher impact on the all record stack.

Line 642: How long were there simulated snapshots?

Line 657: "gasses"

Line 675: So the adjustment is a constant offset over the entire field? Meaning that it doesn't affect the results in any way except shifting the colour scale on the maps a little?

Line 686: I think it would be useful to add a sentence describing what the CCA is about, broadly.

Line 696: Does that then mean that the explained variance on figure 8 are with respect to 60% and 72% instead of 100%?

Line 703: Are the CCA2 really worth interpreting then? It wouldn't hurt to simplify the analysis and figure if possible.

Line 709: My general take on wavelets is that I don't think the quality of the palæo data here is good enough (even though it is indeed high quality pollen data) to estimate time-dependant and timescale-dependant variability estimates at once.

Line 718: Specify the cutoff frequency of the Butterworth filter. Unclear to me why high-pass filtering was necessary for such an analysis. On one hand, one can just ignore the spectral estimates for longer than millennial frequencies if one does not wish to study them, and on the other hand, aren't millennial scale variations better resolved by the reconstructions and more interesting than the more uncertain faster variability? I would perform such analysis without the high-pass filtering and detrending instead, which could be linear detrending or sinusoidal detrending for precession, or no detrending (e.g. <https://doi.org/10.1038/s41561-022-01056-4>).

Supplementary Figures 9-16: Unclear to me why the absolute amplitude of the wavelet power levels between simulations and reconstructions are generally an order of magnitude apart from each other. I think it's because wavelet coefficients are not normalized by the frequency right, so the discrepancy would come from the different resolutions of the input timeseries? In any case, being able to compare the amplitudes would be nice.

Reviewer #2

(Remarks to the Author)

Patterns and drivers of Holocene moisture variability in mid-latitude eastern North America

Submitted to Nature: Communications

This manuscript presents a thorough and creative evaluation of Holocene hydroclimate patterns represented by fossil pollen assemblages and the latest generation of model simulations mid-latitude eastern North America. The manuscript helps address the longstanding disagreement between model simulations and paleoclimate proxy reconstructions over the course of the Holocene in North America. I expect that the findings will be of keen interest to the paleoclimate community. The authors use of a variety of methods and types of data (boosted regression tree proxy calibrations, time slice model simulations, transient model simulations) is a strength of the paper.

The paper would benefit from better organization, especially in the Discussion, as well as a more detailed description of their rationale for carrying out certain analyses and a more thorough explanation of their methodology. I outline the main issues with the Discussion and Methods below. I also include some more minor questions/suggestions for the Figures and for specific lines.

Discussion

I recommend adding sub-headers to the Discussion section to help organize the various pieces of analysis.

A more detailed discussion of how the new BRT-based proxy reconstruction compares to that of the traditional MAT approach would be welcome. Does this study indicate that BRT should be widely adopted instead of MAT?

The authors note that the latest generation of transient climate models do not agree with the reconstructions when it comes to TJul but say that the data-model match is much-improved for the entire growing season heat sum (GDD5) (which is also true for TraCE-21ka). This appears to be truer for the Great Lakes and Northeast region than it is for the Midwest (Figure 2c), which should be acknowledged/explained. Additionally, it would be worthwhile for the authors to describe why it is that the transient simulations produce better agreement for GDD5 than they do for TJul, and why this observation does not hold true for the CESM1 time slice experiments. Finally, is there a reason that the GDD5 values are not included in Supplementary Table 2? (Lines 394 – 402)

The trend reversal in PC2 after 5ka really stands out (Figure 3, Figure 6). Can the authors clarify what drives PC2? Is it the extent of the LIS up until it is gone at ~5 ka at which point the main driver becomes the east-west moisture contrast? Or is it the east-west moisture contrast across the entirety of the record and PC2 aligns so closely with LIS extent from 11 to 5 ka because the LIS extent controls the east-west moisture contrast? (Line 452 – 454)

It is somewhat surprising to the reader that the authors focus the second half of the discussion entirely on the time slice experiments from CESM1. I understand that they focus on CESM1 because they are probing the dynamical relationship between hydroclimate and the LIS, but the authors could do a better job of outlining the rationale for this focus in Lines 456 – 469. Additionally, it would be interesting to see the same dynamical treatment shown in Figures 7 and 8 given to time slices extracted from the transient simulations. These analyses carried out for the transient simulations (individually or as an ensemble) for 8, 7, and/or 6 ka would be useful for further elucidating the influence of the LIS as it disappears. (Lines 490 – 537).

The CCA analysis is confusing and would benefit from a more fleshed-out description. It would be helpful if the authors explicitly stated what the percentages in the Figure 8 sub-headers mean in terms of the CCA.

Methods

What criteria do the authors use to subset the "high-resolution" fossil pollen records?

The authors should outline which model grid cells are used to calculate mean values for each of the three regions. They could also mark the boundaries of these regions (as defined for the spatial extents in the model output) in Figure 1.

The authors should describe how they calculate the water balance synthesis curves in Figure 6.

Figures

Figure 4: I'd recommend showing the points for 12, 11, and 9 ka snapshots from CESM1, and potentially doing away with the lines connecting these points, to underscore that these are not transient simulations.

Why combine MPI-ESM and EC-Earth for this figure while keeping them separate in Figure 2? Does this model ensemble offer something that the individual models do not? The specifics of how this ensemble is calculated should be described in the text. For example, were the models interpolated to the same spatial resolution prior to averaging? Would it be worthwhile to also include TraCE-21ka in Figure 4?

What explains the big difference in sign of the precipitation anomaly between CESM1 at 9 ka and the mean of MPI-ESM1 and EC-Earth at 8 ka? Is this due to differences in the models (CESM1 vs the transient simulations) or interpreted as an actual feature of the climate system?

Can the authors include some metric of uncertainty in this figure, especially for the average of MPI-ESM and EC-Earth.

Figure 7: The authors should consider graphically representing areas where the difference between the time slice experiment (12, 11, or 9 ka) and the PI are not significant. This can be done using a Student's T Test and shown on the figures by graying or whitening out the anomaly (usually around the transition from positive to negative).

Figure 8: I am confused as to what exactly these anomalies are relative to. Can the authors describe how the data in this figure are calculated? Are all these anomalies statistically significant / is there a way to test and communicate this? I understand that CCA1 doesn't always have highest explained variance for P-PET or SLP individually, but is it strange that there are times when the values for both P-PET and SLP are lower than those for CCA2?

Lines

Line 238: What is meant by "a poor spatial coverage of coastal areas in the model."?

Line 353: Add closing parenthesis. Add comma after "In contrast"

Line 432: Can the authors briefly elaborate on the "internal variability" that may be contributing to the ~200-year periodicities that are observed in both reconstructions and simulations?

Line 495: Refer to Figure 8 at end of sentence introducing CCA.

Line 637: Reword the beginning of the sentence about CCSM3.

Reviewer #3

(Remarks to the Author)

This paper presents an impressive synthesis of pollen records and model analyses, leveraging new regression tree methods to attempt to distinguish moisture balance from temperature signals in the eastern US. The boosted regression tree approach appears to provide new insights not available via the common modern analog technique. The authors then compare these results to transient climate model simulations to speculate about the mechanisms responsible for past changes. Overall, I really like this paper, but I think it could benefit from tightening the story and honing the 'big picture' message of the work. I think these could be tackled via major revisions. My overall feeling is that the main strength of this paper comes from the interesting new reconstruction. The model analyses are interesting, but also raise many questions and some confusion in those analyses detracts from the overall strength of this work. I would not be afraid to restructure this paper as a 'data driven' paper.

Below are some overall points to consider as well as detailed comments on the text.

First, I think the organization of the paper would benefit from a combined results and discussion section rather than having these sections distinct. As it stands, there are topics that should traditionally be included in the discussion in the Results, and the organization gets a little bit messy. Especially for a short format paper, combining these would be valuable.

Second, the overall conclusion of the paper could be stronger. The abstract reads more like a literature review rather than

the abstract of a short format paper with a 'punchy' conclusion, and a similar comment could be applied to the paper's discussion. My personal thought is it might be better to focus either on the impact of long-term forcing on the PCA/water balance reconstruction, or on the question of periodicities. As it stands, I don't know that the model analysis is 'enough' to support the conclusion that summer insolation and the Laurentide are driving the patterns you see. But, it is worth noting that Trace21ka has single forcing simulations (e.g. with ONLY orbital changes, or with ONLY ice volume). : <https://www.earthsystemgrid.org/project/trace.html>

It is also worth taking a step back and starting with evaluating the PI model simulations' climatology. Which ones are too rainy over your regions of interest? Too hot? Are the winds too strong relative to observations? This might provide some additional insights about why model performance varies. It would also be good to make a table as to what forcing datasets were used in each set of transient runs - the change in different generations of ice sheet reconstructions (e.g. 5G, 6G, etc) will affect simulations even of the same model.

Finally, there are other 'data' products out there for deglacial climate variables (see Osman et al 2022 in Nature for the data-assimilation approach). My sense is that that product performs well for temperature but might not perform well for hydroclimate-related variables. It would be worth comparing your reconstructions (real world data! So more likely to represent the truth) to the DA product for P and P/ET during the deglaciation. It might help emphasize the innovative nature of the reconstructions that you've produced.

Introduction

Line 126-131: Instead of just reviewing the methods and approaches you used in the paper, provide a signpost of the overall conclusion of the paper.

Results:

Figure 2: Minor point, but I find it hard in some panels to distinguish the lines for EC-Earth and Trace 21ka. Make sure you check colorblindness compatibility of the figure, and perhaps use line weights/dash patterns to increase understandability.

Line 183: This applies to the paper overall, but it is worth defining what you mean by 'drought' - for paleo records like this, we tend to define drought as just below average P or P-E, while the term can mean something very different in the modern literature (e.g. below a certain percentile of precipitation).

Line 196: I think the PCA presents one of the most interesting results in this paper, and it is worth really strongly pinning down what the driver of this pattern is. I am not sure I understand why PC2 is Laurentide deglaciation if it goes 'up' again after 5 ka?

Line 225: I am not sure I agree that there is a good comparison with the evolution of water balance between the models and the data. CESM1 seems to overlap the water balance reconstructions, but seems to be very off in terms of GDD so this to me suggests that it may be getting the 'right' answer in terms of hydroclimate change for the wrong reasons. I am therefore not sure how much to trust the CCA. The logic behind relying on CESM1 for interpretation needs to be clearer. An alternate/complementary analysis might involve comparing Trace to MPI and EC Earth for an interval like 7 ka, when the former is not dry enough and the latter two are too dry. Difference fields between these runs might reveal why the models differ in skill at simulating deglacial climate change. See comments above about some basic diagnostic analyses as well.

Figure 6: In panel d, if this mode reflects Laurentide deglaciation, why does it go back 'up' slightly after 5 ka?

Line 305: I am not sure if the section on periodicities is a good idea for this paper. It seems unrelated to the overall story of the paper, and could be a separate, more robust analysis in and of itself.

Discussion:

Suggest combing with results section, much of this discussion could be integrated through the prior section

Line 394: See comments above about model-data comparison.

Conclusions:

The concluding paragraph of your paper focuses on the MPI, EC Earth, and new reconstructions. If this is the focus, it would be important to then focus on these model simulations in Figures 7 and 8.

I would also check about the length limits for this journal - it seems like there are a lot of figures, and the text is quite long for a short format journal.

Version 1:

Reviewer comments:

Reviewer #1

(Remarks to the Author)

Dear Editor,

I am generally satisfied with the authors' responses and have no major reservations regarding the manuscript. They have addressed my comments in many cases and have carefully justified their choices in others. I appreciate their revisions, particularly the improved methodology for stacking and the clarifications of other methodological aspects.

One aspect that still leaves me somewhat uncertain is the analysis of periodicities. I believe the current framing places too much emphasis on the least informative aspect—wiggles in the wavelet spectra that, after appropriate bandpass filtering, are interpreted as significant periodicities. That said, I appreciate that the authors have tested the sensitivity of their periodicity estimates by varying the null hypothesis, and the analysis is still informative of the distribution of variability across frequencies.

If there are no restrictions on adding new supplementary figures, I would still recommend including the mean wavelet plots used for the periodicity estimates, as these would provide valuable insights. Specifically, plotting the individual wavelet spectra for all sites within a cluster together would allow the reader to see the "periodicities" clustering and, importantly, the overall shape of the wavelet spectra. Alternatively, the mean wavelet spectra could be shown for each region (i.e. a stack of the individual spectra, averaging in the spectral rather than the time domain) with shading to indicate the spread among individual records.

Overall, I see no reason to further delay the publication of the manuscript.

Best regards

Reviewer #2

(Remarks to the Author)

The authors have submitted a response letter and updated version of the manuscript that effectively addresses the comments raised by each of the three reviewers. The reorganization of the Results/Discussion section in particular has strengthened the manuscript. I recommend that the updated version of the manuscript be published and thank the authors for their detailed work to address reviewer comments.

Reviewer #3

(Remarks to the Author)

The authors have largely addressed my comments and those of the other reviewers. I could use some extra clarification on this text near line 428:

"This spatiotemporal pattern has been hypothesized to be explained by LIS deglaciation, and a following waning of the glacial anticyclone and an increased influence of the Bermuda subtropical high, leading to a rerouting of the northward moisture advection towards the NE region and away from the mid-continent"

I'd like a bit more detail - this sentence seems to suggest that after the Laurentide is gone, the upwards trend of PC2 post 5ka could reflect shifts in the strength of the North Atlantic Subtropical High? it would be nice to propose that explicitly in the text.

Given the revised structure of the paper, it would be nice to have a conclusion section that summarizes the overall argument of the paper and a few thoughts on future directions. The last paragraph could be slightly expanded and serve as your Conclusion section.

For the abstract, while it is true that the midHolocene features evaporatively-driven drying similar to the future, it does not necessarily follow that they are analogs, which to me implies a symmetry of underlying mechanisms (and the radiative forcing of CO₂ is different in character than seasonal changes in insolation). Rather, it seems more fair to say that similar processes are at play. I would revise the abstract accordingly.

PATTERNS AND DRIVERS OF HOLOCENE MOISTURE VARIABILITY IN MID-LATITUDE EASTERN NORTH AMERICA

Response to Referees

We quote the reviewers' full comments and describe the changes done to address each point. We are grateful to all reviewers for their thoughtful and detailed comments and we believe that the manuscript has been strengthened by addressing them.

Reviewer 1: *Dear Editor,*

I would like to first apologize to the authors for delays in my submission of the current review. I have read the manuscript from Salonen and al. with great interest and found it interesting and worthy of publication in your journal after revision. While I have reservations on some methodological aspects described below that have to be addressed, I do not think it would fundamentally alter the conclusions of the paper, which rely on several lines of evidence. I have thus a few general comments on the content and structure of the manuscript, and a list of detailed comments with line numbers.

Structure wise, I think that the manuscript could be generally condensed a little both in terms of text and figures. I also note that it seems to me that new results are integrated in the discussion. I don't think the classic results-discussion separation is necessary and that the sections could be restructured with more descriptive titles instead.

We have changed the manuscript to use a combined "Results and Discussion", with thematic subheadings.

The switch to Results and Discussion eliminated about 300 words due to less repetition, and we have mainly used this space to better describe two important elements of the paper: (A) the significance of the difference in data-model match for T_{jul} and GDD5, and (B) the CESM1 equilibrium runs and their CCA analyses at the end of the Results and Discussion.

Reviewer 1: *My general comments have to do mainly with (i) the handling of the reconstructions, specifically relating to the stacking and resolution, and (ii) with the analysis of periodicities.*

(i) If I understand well the stacking methodology, running 250-year windows at 50-year intervals were used to average all the available data points. This means that a high-resolution series with say 5 points in the window would contribute 5 times more than that of a low-resolution series with only 1 point in the window, right? Not to mention that some series might not even contribute to some windows if some of their points are more than 250-year apart. This can lead to apparent non-stationarity in the variability as more estimates are available in the more recent period and thus, on one hand more noise is averaged out, and on the other hand, more smoothing occurs due to time-uncertainty. In both cases, we thus expect less variability (and not more) for stacks of higher resolution series, unless the signal-to-noise ratio is high and the time-uncertainty is low. I would thus suggest to at least perform a temporal interpolation before averaging the series as was done for example by <https://doi.org/10.1038/nature25464>.

Thank you for this and other comments; we appreciate the careful review. We have done the temporal interpolation of the individual site-specific reconstructions as requested by Reviewer 1, using a 50-year time step.

Importantly, we note that in our original submission, we *already* used the data with interpolated values at 50-year time steps for both the PCA and the wavelet analyses. Thus, the PCA and wavelet analyses are entirely unaffected by the new synthesis curve methodology.

Using the equivalent interpolated data also for the stacked reconstructions thus overall simplifies the methodology of the paper.

With the new stacked reconstructions based on the interpolated site reconstructions, we continue to use bootstrapping to calculate the 95% errors for the ensemble mean, using the values interpolated for the individual sequences for each time point. To summarize multi-centennial trends, we also plot the running 5-point mean of the ensemble means.

From a methodological angle, we agree with Reviewer 1 that the benefit of the interpolation of the individual sequences is that it equalizes the impact of higher and lower resolution fossil sites on the synthesis curve. This can be desirable especially in the study of *millennial* trends, because (A) even the lower-resolution sites should be able to resolve the millennial trends and (B) because individual sites may be subject to persistent biases in pollen taxon percentages due to taphonomic factors such as basin size. When considering *centennial* features, one might counter that ultimately a higher-resolution sequence provides more observations on fossil pollen variation and should be given proportionally more weight. (Anecdotally, in another ongoing study we have faced a situation where interpolating the data sequences destroys a rather beautiful record of the 8.2k event provided by the highest-resolution fossil sites.) However, because our largest focus here is on millennial spatiotemporal patterns, and because our Wavelet analyses used to study periodicities employ the individual sequences anyway (instead of stacked reconstructions), on balance we do see the benefit of the interpolation for this study.

The figure below compares the new vs. old reconstruction synthesis curves:

OLD: 250-yr moving average of reconstructions (*dashed lines*)
NEW: Mean of reconstructions interpolated at 50-yr time step + 5-pt mean (*solid lines*)

To summarize, these are the major changes in the synthesis curves:

- Overall, the synthesis curves do not change greatly, and there are no changes to key interpretations of the paper.

- Some changes are seen in the millennial moisture levels, especially in MW and NE where the use of the subsets of only 6 high-resolution pollen sequences makes the reconstructed moisture level sensitive to cases where some comparatively high-resolution sequence falls towards the edges of the ensemble. In particular, a deepening early-Holocene midcontinent-coastal contrast is seen due to wetter conditions in MW and a deeper drought in NE. This reinforces one of the major observations in the paper.
- Over the past 8 ka, the fit between water balance reconstructions and simulations (shown in the revised Fig. 2) improves further for both GL and NE, with GL showing deeper mid-Holocene drought, and NE somewhat shallower drought, which agrees with the new-generation transient models (EC-Earth and MPI-ESM).

We have also recalculated the synthesis curves of the transient climate simulations shown in Fig. 2, using 50-year binning of the annual values instead of the 250-year binning used in our original submission, to remain as consistent as possible with our new reconstruction synthesis curves. This change mainly affects the width of the 95% error bands and is not important for the paper.

Reviewer 1: *Is there evidence that BRT improves the signal-to-noise ratio (SNR) within clusters compared to MAT? It seems clear that the variability of the BRT-based stacks is higher than the MAT-based ones, but is this due to higher SNR with BRT, or simply because the variance of BRT reconstructions are generally higher than that of MAT ones? In other words, is the variance ratio from individual series in clusters to their stack the same for both methods, or is it really higher (lower) for MAT (BRT)?*

Our choice of BRT for this paper draws on the cross-validation (CV) experiment shown in Salonen et al. (2019) which compared eight calibration models (including BRT and MAT) which uses a robust h -block CV scheme, indicating BRT as the best-performing model. That paper further showed that the BRT performance advantage over MAT would further increase with very high h values, which mimics the situation faced in paleo-reconstructions with poor modern pollen analogs.

We acknowledge that even this approach cannot fully represent the problem of reconstruction from fossil data, where we suspect the reliability of methods remains, to a significant extent, unknowable. Hence we decided to also include the full MAT results to show that the main conclusions of the paper are not sensitive to calibration method choice.

From a practitioner viewpoint, our preference for BRT is also affected by the exceptionally good tools BRT provides to verify that the paleo-botanical signals (i.e. variation of specific pollen types) used to reconstruct a given climate variable is consistent with ecological knowledge. In this study, such analysis of the BRT models gives strong ecological validation for the BRT calibration models (Supplementary Figs. 12 and 13).

We have edited the reconstruction section of Methods to more clearly state this reasoning.

Reviewer 1: *(ii) While the analysis of periodicities seems to be computationally correct, I have reserves on the methodological choices. The methods used have however been used quite frequently in palaeoclimate studies and therefore my criticism is not necessarily specific to the author's work. First, I have reserves on the usage of wavelets to obtain time-dependant and timescale-dependent variability estimates from palaeoclimate data, the issue being that we are extracting a lot of information from relatively short and noisy timeseries, and changes in periodicity through time is likely random rather than expressing a real change in the state. For such Holocene timeseries, resolving about one order of magnitude in frequencies, more robust results would be obtained by looking at the wavelet (or regular power) spectrum for the whole time period (e.g. Fig.1 in <https://doi.org/10.1038/s41561-022-01056-4>).*

We calculated the wavelet for the past 8 ka, the same as in Hébert et al. (2022). We just removed the Early Holocene because of the influence of the LIS in the lakes.

Reviewer 1: *Wavelet spectrogram can still be qualitatively informative in this case to see how the variability is distributed across time and frequency, but to infer quantitative information requires some additional considerations. The null hypothesis has to describe the “background variability” correctly. While an AR1 red noise is commonly implemented and used in palaeoclimate analysis, it almost never describe the background variability satisfactorily and leads to spurious estimation of peak significance (e.g. <https://doi.org/10.1175/JCLI-D-22-0011.1>). Therefore, most, if not all, of the periodicities are likely not significant (and with $p < 0.05$ we still expect 5% of false positives). I also note here that I wonder how the periodicities were identified from the spectrogram as they are not stationary time, was that from a mean wavelet spectrum?*

The periodicities are indeed from the mean wavelet spectrum.

It is generally difficult, and for specific cases almost impossible, to know the “true” background variability which makes AR1 the most plausible choice. A better grip on the color of proxy noise might help in future studies (e.g. McPartland et al. (preprint): <https://cp.copernicus.org/preprints/cp-2024-73/>). Using many records as done in this study has the advantage that we can be more confident about the robustness of the results if local records consistently yield comparable frequencies independent of the type of significance testing.

Out of curiosity, we tested 44 different methods/parameter choices (described below) of describing the “background variability” and, therefore, to calculate the significance of the observed periodicities. Most of the periodicities were robust enough that they appear in most, if not all, the methods. The tested methods included:

- White noise; $N = 1$
- "shuffle" (shuffling the given time series); $N = 1$
- “Fourier.rand” (time series with a similar spectrum); $N = 1$
- “AR” (AutoRegression), with “p” (the number of past observations required to define the auto regression) varying between 1 and 5; $N = 5$
- “ARIMA” (AutoRegressive Integrated Moving Average), with “p” (the number of past observations required to define the auto regression) varying between 0 and 5, and “q” (the number of past observations required in the moving average) also varying between 0 and 5; $N = 36$

However, red noise (AR1) is usually implemented in time series analysis because its power spectrum is weighted toward low frequencies. Therefore, for the manuscript, we decided to use only the tests based on red-noise methods, i.e., AR-1, ARIMA(101), ARIMA (102), ARIMA (103), ARIMA (104) and ARIMA (105). For the final figure in the manuscript, we kept only the periodicities that were significant in more than 66% of the tested methods (i.e., in at least 4 out of 6 AR-based methods).

We have revised the Supplementary Table 3 to show all periodicities observed originally using AR-1 method, and in addition, how many times (in %) the peaks were significant considering the 6 AR-based tested methods. From the results shown in Fig. 6, we have then filtered out the periodicities which were significant in three or fewer of the tested methods. This results in relatively minor changes in the kernel density plots fitted to the periodicities, and doesn’t impact our main conclusions about the periodicities.

Reviewer 1: *I also have reserves with respect to the frequencies that could be resolved given the low- and high-pass filters that were applied. The pollen data is computed with 250-year sliding windows, which corresponds to a convolution with a rectangular window, or with the multiplication of a sinc function in the Fourier domain. What this means really, is that the cutoff frequency affected by the sliding window, effectively a not so ideal lowpass filter, is greater than $1/(500 \text{ years})$. Then on top of that, a high-pass filter is used to kill the power below $1/(1000 \text{ years})$, and thus we have a very narrow window between $1/500$ and $1/1000$ years where variability estimates are less affected by the smoothings (albeit the*

running mean filter has a sizeable impact even on the (1/1000 years) frequency). What is then the sense of the AR1 null hypothesis when the shape of the spectrum does not represent the signal but the smoothing filters right? In the end it looks like the identified periodicities are quite randomly distributed over the range of frequencies and I'd suspect that any similarity, albeit I fail to see the described broad agreement, would be more likely an interaction between the shape of the filters and that of the null hypothesis.

First, we note an apparent misunderstanding represented by the statement “the pollen data is computed with 250-year sliding window” and we apologize for lack of clarity in the original submission. We did indeed perform our multi-site synthesis reconstructions in our original submission with the 250-year rolling window – although this approach was changed for this revision, as discussed above. However, the wavelet analysis was not performed on these synthesis curves, but on the individual, underlying pollen-based reconstructions, and even then, specifically *only* on the 12 highest-resolution reconstructions available for the Midwest and Northeast clusters. As we note in the periodicity subsection of Results and Discussion, these pollen sites have an average time resolution of about 60–70 years. These reconstructions had been further interpolated at 50-year time step prior to the wavelet analyses; as we note above, this was also done for the data incorporated in the synthesis curves in this revision.

We have edited the Methods section about the wavelet analysis, to be clearer on the subset of data used for the analysis.

Reviewer 1: *I think that in order to evaluate whether the water balance and July Temperature are linked, I would suggest a cross-spectral analysis. If common periodicities do exist, they should be clearly evidenced by such analysis as significant levels of coherency should be attained at the relevant frequencies.*

Our main focus was to identify if the water balance and the July temperature presented significant periodicities. Because our intent was not to test if they varied simultaneously, we prefer to keep the wavelet analysis of the two parameters separate. Additionally, a cross-spectral analysis would add even more discussion to an already very complex paper.

Reviewer 1: *Below are detailed comments with line and figure references:*

Line 72-76: A bit of a weird sentence, you say MAT and responses surfaces were used before, and that not MAT and other classical methods are used. I'm not sure what the logic is, except to say that response surfaces might not be used so much anymore? I just think the sentence could be simplified to say "different methods such as X and Y have been developed and used."

Line 78: Last two sentences could be combined and shortened, and a summary sentence could be added to the paragraph with the key takeaway point.

We have edited this section, with a simplified list of reconstruction approaches as suggested by the reviewer, and in the following sentence clarified our intended meaning, i.e. contrasting the quantitative moisture reconstructions with indirect measures of moisture variability.

Reviewer 1: *Line 82: Is it really abrupt if it occurs over a millennium? Abrupt kinda implies a departure from more "normal" variability to something more extreme, I'm not sure if that's really the case, at least not when we look at the average of several reconstructions (which might be overly smoothed) but on the other hand single reconstructions are likely containing lots of non-climatic noise.*

We agree the word “rapid” is not needed here; removed.

Reviewer 1: *Line 94: The time is always ripe for a new and better analysis. I think a "Why" could be included here too already, "because of new methods and data".*

We have revised the opening sentence here, stating more plainly that we're here preparing a new synthesis and analysis of the eastern North American hydroclimate (because of these new advances which we then describe).

Reviewer 1: *Line 95-96: "have arisen" sounds like they just appeared on their own, they were developed (by people).*

Rephrased → "have been developed".

Reviewer 1: *Line 118: transient climate model simulations? I would maybe use the phrasing: "an ensemble of N Holocene transient simulations across 4 climate models"*

Rephrased → "an ensemble of four climate model simulations for the Holocene".

Reviewer 1: *Line 125: No need to repeat there are four sets of model simulations here, just "the Holocene simulations"*

Done.

Reviewer 1: *Line 128: Finally instead Fourth?*

Done.

Reviewer 1: *Line 129: Maybe skip the "concerning the spatiotemporal patterns.... America" and just say directly "we test the hypotheses that the patterns are linked to..."*

Done.

Reviewer 1: *Figure 1: Just wondering what the reason is for the lack of records between the Great Lakes cluster and the Northeast cluster? Lack of high resolution records for these areas or because the signal is too mixed there and not useful for the analysis? I would also suggest to move the map to the extended data figures.*

The cluster limits reflect the (rather long) development history of this paper. We were originally working with the Midwest cluster only, representing the much-studied forest-prairie ecotone and a real clustering of fossil sites. After being satisfied that the BRT water balance models are also useful within the wetter forest ecosystems, we introduced the Northeast cluster which also had a real concentration of published high-resolution pollen data.

The Great Lakes cluster was the last addition. Here, the gap eastward before reaching the Northeast cluster is indeed real – after filtering the fossil pollen sites available on Neotoma for our required pollen sample and dating counts, no data were available to expand the Great Lakes cluster further eastward.

We agree Fig. 1 could potentially be in the Supplement. But given the relaxed article format of *Nature Communications*, within which we also comfortably fit, we prefer to keep the map figure, as it nicely grounds the introductory paragraphs about North American environmental history and the data used in this paper. Note that we have now also added boxes in the map used for calculating the data regions from climate models (see another reply below, p. 17).

Reviewer 1: *Line 150: Typo "the MD" maybe "in the MD"?*

Fixed.

Reviewer 1: *Line 151: Typo "at a 10-7 ka"*

Fixed.

Reviewer 1: *Line 159: Unclear what means that the progress of the drought is well tracked by proxy data. Do the authors simply mean that negative water balance anomalies are reconstructed for all regions from 10ka onward?*

Rephrased to say that we mean the lack of specific biases due to poor modern analogs.

Reviewer 1: *Figure 2: Wondering if there are ways to simplify the figure and caption. Could we show the average of MPI-ESM and EC-Earth together and maybe only show MAT in the extended data fig? Please indicate the bandwidth h with units and without the logarithm on the axis (the spacing can remain logarithmic, but the labels would be clearer if given directly in ka). On the other hand, are those SiZer maps really necessary? What do they tell us except what we can already see on the plot? The GDD5 is maybe not necessary to be shown? Was there an additional CESM1 pre-industrial snapshot that was considered to calculate the anomalies? Could the models be combined as on Fig 4, and maybe keep TraCE-21ka for the supplement?*

We broadly agree with the above, and have done significant simplifications to Fig. 2:

- The MAT reconstructions have been moved to supplement (new Supplementary Figure 16)
- The GDD5 reconstructions have also been moved to supplement (new Supplementary Figure 11).
- We have done the suggested change to the SiZer panels, showing the smoothing bandwidth directly in ka. We have, however, opted to keep the SiZer maps in the figure. Methodologically, we're quite fond of the SiZer because it simultaneously provides an assessment of significant trends on different timescales, while doing smoothing at a full range of bandwidths and without requiring a subjective decision on the bandwidth to use (as is the case with e.g. LOESS).

With the overcrowding of the figure now greatly eased, we have opted to keep the transient models separated.

About CESM1, there was indeed a separate PI control run; the results from this run are shown in Supplementary Table 2. We have added a note about the CESM1 PI run in the Fig. 2 caption.

Reviewer 1: *Figure 2: Presumably the temperature and water balance ratio are relatively well correlated in the model simulations. Therefore, if water balance agrees with the reconstructions but not July temperature, I wonder whether a model that would simulate correctly the July temperature would then end up overestimating the water balance?*

The water balance integrates different aspects of being a function of annual precipitation and potential evapotranspiration (PET). PET is generally a function of temperatures above freezing point but includes both, how warm it is overall and how long the warm season is. As P and PET may have compensating effects or not, the correlation is not necessarily obvious or stationary. See also in another reply further below (p. 22) and how P vs. PET differs over time in Fig. 4.

Reviewer 1: *Line 190: Can the details of the comparison between BRT and MAT be moved to the methods or supplement to improve the readability flow?*

Agreed. We have moved the BRT-vs-MAT comparison away from the main text. There is a new Supplementary Figure 16 comparing the BRT and MAT reconstructions. We run through the differences to MAT in the Methods, after presenting the BRT reconstruction approach.

Reviewer 1: *Line 196: Maybe specify that the PCA is done across all the clusters.*

Done.

Reviewer 1: *Line 198: Maybe write "with the remaining components not exceeding 6.1%" rather than using a symbol " $< 6.1\%$ ".*

Done.

Reviewer 1: *Line 202-207: Could the description of PC2 be better and more simply explained? I just wonder a bit, what is the longitudinal water balance contrast, or the period of maximum contrasts due to peak multi-millennial drought. I think this could be said more simply. I don't know if the PC loadings can be said to be moving with longitudes, I see mostly two clusters with positive loadings in the interior and negative on the coast, and not so much transition in between.*

Line 209: Are the two loadings covarying with the midwest sites doing so significantly? Given that the original series might have different resolutions (and I imagine the PCA is performed after interpolation), high amplitude loadings could still be random or dominated by noise.

We agree that the PC2 loadings are better described as positive interior vs. negative coastal clusters. We have significantly edited this paragraph, to simplify the statement about the spatial pattern in loadings. We have also rearranged the wording to *first* present the site loadings and thus the interpretation of each PC, and only then describe the temporal changes in sample scores. The panels in the associated Fig. 3 have been reordered accordingly.

Reviewer 1: *Line 209: Are the two loadings covarying with the midwest sites doing so significantly? Given that the original series might have different resolutions (and I imagine the PCA is performed after interpolation), high amplitude loadings could still be random or dominated by noise.*

Whether a higher order PCA mode is meaningful is based on the explained variance of that mode and whether its interpretation can be linked to a "real" phenomenon. Based on the eigenvalues representing the explained variances in Figure 3a, this is clearly the case for the first two modes. Apart from this, PCA loadings are typically not tested for statistical significance as other statistical measures. The loadings only say how strong the represented variability is within a given mode and the sign is used to indicate co-variability in phase or anti-phase. The lower the specific loadings, the less important within a given mode. Overall, PC1 and PC2 explain a significant amount of the overall variability and can be physically explained as in Fig. 5

Reviewer 1: *Figure 3: What does "scores for each sample of water balance reconstructions" mean? Revise and clarify sentence.*

We have significantly clarified the Fig. 3 caption.

Reviewer 1: *Line 272: The shift from negative to positive PET anomalies looks like model bias, it's doubtful that a continuous curve could go through. Are there no simulations of CESM1 that overlap with the other runs to evaluate this? Again I wonder about how the CESM1 anomalies were computer and whether this could be related.*

There are two main problems here:

1. Unfortunately, there are no transient simulations with sufficiently high resolution connecting the early Holocene with the rest of the Holocene. While TraCE simulations do capture the whole period, the very coarse spatial resolution and potentially other reasons

lead to fundamentally inconsistent results of that model with other models and proxies. We now highlight the use of different models by splitting up Fig. 4 for the change in models.

2. The transient high-resolution simulations of MPI-ESM and EC Earth starting at 8 ka BP start from initial conditions that do not include any memory in the ocean from coming out of the ice age. This might explain the big jump relative to CESM1, however, there might also be other reasons that could only be resolved with more transient simulations for the whole period. For the specific region, the switch from still having an ice sheet (CESM1) to its demise (other models) may further contribute to the jump. The main motivation to use the set of models included in the paper is to highlight the role of ice sheets (CESM1) in the early Holocene and the transient evolution afterwards (other models).

Reviewer 1: *Figure 4: It's interesting to note that the CESM1 simulations show a wetter midwest while the transient ones show a wetter northeast. I wonder what we learn from the difference regional curves here other than this reversal between the models which doesn't seem to be interpreted. If we cannot trust the regional differences from the simulations, I would show instead one curve for the average across all three regions. Again, it would be interesting if an addition CESM1 simulation at 8ka and thus overlapping with the transient simulations was performed in order to see if the discrepancy is a model bias. I think the figure could be improved in general better evidence the key takeaway.*

Based on the proxy reconstructions, the water balance should differ regionally in the early Holocene as simulated by CESM1 in response to ice sheets. This west-east gradient changes with the disappearance of the ice sheet in the proxies consistent with the transient climate models starting at 8 ka. So the difference between CESM1 and the other models are real changes consistent with the proxy data and are not dominated by the switch of models. Please note that the model data is presented as anomalies relative to the pre-industrial simulation of each model. Hence, systematic differences between models are largely already removed by this approach.

Reviewer 1: *Figure 5: Revise figure. Either show the average spectra or the coherency to show common periodicities.*

The coloured (dashed and dotted) vertical lines in Fig. 5 (Fig. 6 in our revision) indicate the mean wavelet spectrum of each dataset. Because of the amount of analysed data (2 regions with 6 lake records and 3 model simulations per region, and 2 climate variables from each dataset or model), we decided to present all observed significant periodicities as the vertical lines and integrate them using a Kernel density plot. Presenting each average spectra individually would represent adding a panel made of 36 different plots, which would be even more difficult to identify similar significant periodicities.

Reviewer 1: *Line 344: Yet to prove with a coherency analysis.*

See other reply below (p. 10); we have softened the statement on the coherency of the centennial events.

Reviewer 1: *Line 353: Missing closing bracket.*

Fixed.

Reviewer 1: *Line 362: Again unsure about the periodicity argument.*

This passage was removed in combining the Results and Discussion.

Reviewer 1: *Figure 6: It seems that all the data shown here has been shown before on different panels, it would be nice if somehow we could condense the storyline and avoid*

duplication. (a) Seems like the Great Lakes and Northeast might have some anti-correlation, might show up on coherency analysis. (b) Why the shading under the curve? What high-pass filtering was done? Looks like the cutoff frequency might be too high, I would favour only detrending the very slow trend (could be done by subtracting a loess smoothing with a long span (on the order of a couple thousands of years). (c) Seems that the deglaciation could explain the first part of both curves, and that the second part would be the insolation trend.

We would favor keeping this figure. First, we think the summary of water balance patterns in all clusters in a single panel is helpful for the Discussion and to summarize the paper. Also the centennial variability and the comparisons vs. forcings do add significant new results.

We have done the requested change to the method of calculating centennial variability, using the residuals of a 2-ka LOESS smoother.

Notably, with the changes both to the method used to analyse the centennial variability and to the underlying climate curves (due to the new methodology described above), the results do change quite significantly:

- The 8.2k event doesn't stand out as clearly, and we have removed to discussion regarding this event.
- The 5.5k wet shift still stands out as a centennial feature shared by all clusters, and the discussion regarding this is retained. However we now note the coherency of the 5.5k event between regions could well be coincidental.
- There's no longer an evident shift in the degree of centennial variability over the Holocene. We note this briefly in the revised manuscript.

The disappearance of known events like 8.2k due to using a different filtering or interpolation method highlights that these choices should depend on the research question.

In sum, we are significantly shortening the discussion on centennial variability. This also gives us space to expand the discussion of the drivers of millennial change in a later part of the Results and Discussion (see below).

We have opted to not do the suggested coherency analyses. With the overall direction being towards expanding the discussion of the drivers of millennial change (and better discussion of Figs. 7 and 8) – while staying within *Nature Communication's* length limits – we favor not expanding but rather slightly shortening the discussion on centennial variability.

See comment below (p. 16) about the interpretation of the PC1 and PC2.

Reviewer 1: *Line 387: How were the climatic impact of vertical land movements estimated? Why 0.5 K? Unclear to me how vertical land movements affect the climate.*

Line 388: Why are they expected to negatively bias the HTM amplitude and delay reconstructed HTM?

The idea here is that because of the isostatic readjustment, our proxy sites don't capture a paleoclimate record from a stable elevation. Effectively, the isostatic forces carry the paleothermometer from one elevation to another over the millennia, and hence, the reconstruction doesn't represent the climate change that would have been observed at a site of stable elevation.

Relevant for this work, in a region of forebulge collapse, the sites would have been at increasingly higher elevation towards the early Holocene, leading to an increasing negative bias in reconstructed temperature towards the early Holocene.

Hence, in a region of forebulge collapse, the site would have also been at a higher elevation during a mid-Holocene temperature maximum, making the reconstructed temperature amplitude vs. preindustrial lower than would've been the case at constant elevation. Also, because the negative bias in temperature increases towards the early Holocene, this delays the arrival to the mid-Holocene temperature peak.

This quick sketch perhaps helps illustrate the idea:

We can estimate the maximum bias taking the lapse rate of $6.4\text{ }^{\circ}\text{C}/\text{km}$ and the magnitude of forebulge collapse in the Great Lakes region, which can be visually estimated at slightly under 100 m based on a figure in the Clark et al. (2012) paper we cite. We thus come to a rough estimate of $0.6\text{ }^{\circ}\text{C}$ for the maximum bias – we have changed this estimate from the rougher estimate “ $\sim 0.5\text{ }^{\circ}\text{C}$ ” given in our original submission, based on the cited lapse rate.

We have done multiple edits to this paragraph which hopefully clarify our reasoning.

Reviewer 1: *Line 397: What is the contradiction?*

We have significantly expanded this paragraph to explain the implications of the proxy-data match achieved for T_{Jul} and GDD5.

Reviewer 1: *Line 399: So are you suggesting that the July temperature reconstructions are of poor quality? Then why use them at all?*

“Poor quality” is perhaps a harsh way of putting it, but we do consider that for a vegetation-based proxy a variable like GDD5 encompassing more of the growing season might be more optimal. In practice, however, we’re often stuck with T_{Jul} /MTWA to enable synthesis with other proxies and climate modelling, and also to allow comparison with the plethora of earlier studies working with T_{Jul} /MTWA. The problem of using peak warm or cold temperatures goes beyond the current study and touches upon the general problem of potential seasonal biases, i.e., if changes in seasonality and/or its effect is not known.

Reviewer 1: *Line 404: What is meant by "significant" variations? Significant should be used in the context of testing.*

This sentence was removed in combining the Results and Discussion; however this referred to the significant deviations of the derivative from zero, as determined by the SiZer analysis.

Reviewer 1: *Line 405: Are they however coherent between regions? I imagine that there is a high level on noise on single reconstructions, but still wondering on the robustness of centennial scale variations.*

This was a miswording and meant to say “coherent between regions”. This is corrected in the revised section on sub-millennial variations.

Reviewer 1: *Line 412: Could the reason that the 5.5-5.0 ka wettening is visible across the three regions simply that it has higher amplitude and thus creates a stronger signal above the noise in all three regions, whereas elsewhere the centennial variability is masked by proxy noise?*

With the new synthesis curves, we have softened the statement on the 5.5k event (see reply above, p. 10).

Reviewer 1: *Line 424: Were the series first interpolated before the average? The higher number of samples in the recent part means that more noise will be averaged out and therefore, the difference in variability might be an indication that more proxy noise remains in the earlier part. In this sense, the higher data resolution should average more noise and decrease variability.*

As noted, we have revised the synthesis curves to use interpolated data.

Reviewer 1: *Line 429: 200-year is below the effective resolution of the series, which were produce with 250-year sliding windows. It does not make sense to interpret that timescale. Line 430: Or maybe the significant periodicities around the 200-year frequencies correspond to a transition timescale of the AR1 null hypothesis where a mismatch is more likely to occur and lead to spuriously significant periodicities? That would be my guess.*

See reply above (p. 5) about the nature of the data used for the wavelet analysis.

Reviewer 1: *Line 474: Period missing.*

Fixed.

Reviewer 1: *Line 493: What is meant by comparing large scale SLP and regional water balance? Do I understand right that in one case the CCA is performed over a region from longitude 170W to 40W and Latitude 20N to 75N and the other 100W to 40W and 32N to 58N only over land? These seem to be very arbitrary choices to me. Of course this type of analysis can be sensitive to the data that is included and choosing different domains can change the dipole spatial patterns observed.*

We described the CCA in more detail in the methods sections which, however, comes only at the end of the manuscript. We now add two sentences to better describe the rationale of using the CCA and the choice of the regions in the revision version already briefly in the Results and Discussion and refer for details to the Methods. As written in the latter, in general, CCA finds pairs of linear combinations (canonical variates) from the two sets of variables that are maximally correlated with each other.

In our case, we use CCA to find the modes of maximum correlation between a local variable – here the small domain of the water balance where the proxy sites are located – with the large-scale atmospheric driver here represented by continental-scale SLP over whole North America. So the domain of the water balance has to be local to be comparable with the proxies. Otherwise, the CCA would no longer explain the local variability of the target region of the small domain. In general, the selection of the domains for the CCA is always somewhat arbitrary, but we tried to account for the large-scale character of SLP changes using a larger domain and the local-scale character of the water balance using a smaller domain. The smaller domain was specifically selected to appreciate the concrete location of the proxy sites used in the study.

Reviewer 1: *Figure 8: Are we really learning something from all those figures? Could it be simplified? I am not so familiar with the details of CCA, but I wonder if the correlations are significant? I think the Methods on CCA could better describe the procedure.*

We simplified the layout of the figure and enhanced the explanation in the text as mentioned above. Despite the noisy nature of water balance, the correlations with the large-scale SLP variability are generally quite high as shown by the numbers in Fig. 8. Interestingly, the correlations are systematically higher in the early Holocene ($r > 0.4$) than during the PI, possibly due to the role of ice sheets in the past which is absent today. We will add to the figure caption that all correlations are statistically significant at the 99% level (e.g. the smallest $r = 0.34$ in CCA2 for PI with $p < 0.01$ ($p = 0.00053825$)).

Reviewer 1: *Line 539: I struggle with this statement given that the rise in PET from 9ka to 8ka in model simulations was inferred from a change from CESM1 to transient simulations.*

See another reply above (p. 8). The general switch from low PET in the early Holocene results from still having an ice sheet and hence cooler temperatures nearby. This effect is gone when the transient models start at 8 ka BP.

Reviewer 1: *Line 616: Using 250-year windows means that the centennial scale variations, which were analyzed and interpreted, will be very smoothed. I think any variations below that should not be interpreted, at least in terms of their amplitude as they will be damped by construction. I think it would be better to interpolate to 50-year resolution all the series and stack them. The series could then be smoothed when plotting timeseries to evidence slower variability, but unsmoothed series should be used for wavelet/spectral analysis.*

As noted above (p. 2), in the revision we have done the interpolation of the individual sequences to 50-year resolution before stacking the reconstructions.

Reviewer 1: Methods
Line 572: comma before "which".

Done.

Reviewer 1: *Line 573: Split sentence before "however".*

Done.

Reviewer 1: *Line 584: How were the cluster limits chosen? I wonder whether the gap between the GL and NE areas are a choice or a lack of records.*

Addressed above (p. 6).

Reviewer 1: *Line 587: Well-dated is good, but what does that mean concretely in terms of time uncertainty? Generally, I'd still expect a time uncertainty on the order of hundreds of years, meaning that stacking nearby sites would lead to a smoothing of the signal due to 'misalignment'. The effect can be seen on surrogate data created from model simulations with realistic time uncertainty and the resolution of pollen records on Ext. Data Fig 1 in <https://doi.org/10.1038/s41561-022-01056-4> . Of course the effect should be smaller in this case given the selection for high quality records, but I still suspect it to be important for the centennial variability. I also wonder here the impact of sediment mixing on the effective resolution of the high-resolution records*

We agree on all counts. Overall, our revised manuscript deals somewhat less with the sub-millennial features seen in the stacked synthesis reconstructions. Meanwhile, as noted above (p. 5), the analysis of periodicities doesn't use the stacked reconstructions at all, but the individual reconstructions from the subset of high-resolution sites.

Reviewer 1: *Line 616: Were the reconstructions interpolated before taking the rolling mean, as was done for example by Marsicek et al 2018, albeit here there would be no spatial gridding since we are only interested in stacking the clusters.*

Line 625: Well it wouldn't be suprising that the stack of all records and high-resolution records show similar features as they are both using the high-resolution data, and if there is not interpolation before stacking done, then the high-resolution one will contribute more samples to the rolling mean and have a higher impact on the all record stack.

Addressed above (p. 2); now doing the interpolation of reconstructions before stacking.

Reviewer 1: *Line 642: How long were there simulated snapshots?*

As described in Schenk et al. (2018) and Kuang et al. (2012), the simulations were run for 150 years but only the last 100 years are used for analysis. We have added this to the Methods text.

Reviewer 1: *Line 657: "gasses"*

Fixed → "gases".

Reviewer 1: *Line 675: So the adjustment is a constant offset over the entire field? Meaning that it doesn't affect the results in any way except shifting the colour scale on the maps a little?*

Yes, subtracting the systematic model differences (offset) in global mean sea-level pressure between past and pre-industrial climate state experiments helps avoid spurious results. For accurate dynamical interpretation and comparison of surface or sea-level pressure, it is essential that the surface or sea-level heights are consistent. Given the significant changes in sea levels and ice sheets in the past which are incorporated in CESM1, removing the systematic offset is necessary to achieve this consistency.

Reviewer 1: *Line 686: I think it would be useful to add a sentence describing what the CCA is about, broadly.*

We have added this in the revised version.

Reviewer 1: *Line 696: Does that then mean that the explained variance on figure 8 are with respect to 60% and 72% instead of 100%?*

The explained variances in Fig. 8 are calculated relative to 100% of the input data used in the Canonical Correlation Analysis (CCA). While the input data for the CCA represents a subset of the total simulation data (due to the removal of "noise" by the Empirical Orthogonal Functions (EOFs)), the variances shown in Fig. 8 are based on the proportion of variance explained within this subset. Therefore, the percentages (60% and 72%) reflect the explained variance relative to the input data used in the analysis, not the total amount of variance represented by the original simulation data.

Reviewer 1: *Line 703: Are the CCA2 really worth interpreting then? It wouldn't hurt to simplify the analysis and figure if possible.*

The high canonical correlations and large explained variances of CCA2 clearly require it to be included. As written in the text, it is also physically-dynamically required because CCA1 reflects the dominating large-scale dominance of pressure variations over the ice sheet while CCA2 reflects the dependence on the strength and position of zonal pressure gradients.

Reviewer 1: *Line 709: My general take on wavelets is that I don't think the quality of the palaeo data here is good enough (even though it is indeed high quality pollen data) to estimate time-dependant and timescale-dependant variability estimates at once.*

See reply above (p. 5) about the nature of the data used for the wavelet analysis.

Reviewer 1: *Line 718: Specify the cutoff frequency of the Butterworth filter. Unclear to me why high-pass filtering was necessary for such an analysis. On one hand, one can just ignore the spectral estimates for longer than millennial frequencies if one does not wish to study them, and on the other hand, aren't millennial scale variations better resolved by the reconstructions and more interesting than the more uncertain faster variability? I would perform such analysis without the high-pass filtering and detrending instead, which could be linear detrending or sinusoidal detrending for precession, or no detrending (e.g. <https://doi.org/10.1038/s41561-022-01056-4>).*

The cutoff frequency was 1/1000 years, to remove the multi-millennial frequency. We have added this information to the manuscript Methods.

The high-pass filter was necessary because the magnitude of the millennial time scale is disproportionately larger than the sub-millennial time scale. We tested running the wavelet analysis without removing the multi-millennial frequency, but all the sub-millennial oscillations were masked. Therefore, to be able to resolve the not so well studied sub-millennial oscillations, we decided to apply the high-pass Butterworth filter.

Reviewer 1: *Supplementary Figures 9-16: Unclear to me why the absolute amplitude of the wavelet power levels between simulations and reconstructions are generally an order of magnitude apart from each other. I think it's because wavelet coefficients are not normalized by the frequency right, so the discrepancy would come from the different resolutions of the input timeseries? In any case, being able to compare the amplitudes would be nice.*

Indeed, the wavelet coefficients were not normalized by the data frequency. The pollen data has a 50-year resolution, while the model simulations' data has a 1-year resolution. This difference magnifies the amplitude of the wavelet power levels from the model data when compared to the pollen data.

Reviewer 2: *This manuscript presents a thorough and creative evaluation of Holocene hydroclimate patterns represented by fossil pollen assemblages and the latest generation of model simulations mid-latitude eastern North America. The manuscript helps address the longstanding disagreement between model simulations and paleoclimate proxy reconstructions over the course of the Holocene in North America. I expect that the findings will be of keen interest to the paleoclimate community. The authors use of a variety of methods and types of data (boosted regression tree proxy calibrations, time slice model simulations, transient model simulations) is a strength of the paper.*

The paper would benefit from better organization, especially in the Discussion, as well as a more detailed description of their rationale for carrying out certain analyses and a more thorough explanation of their methodology. I outline the main issues with the Discussion and Methods below. I also include some more minor questions/suggestions for the Figures and for specific lines.

Discussion

I recommend adding sub-headers to the Discussion section to help organize the various pieces of analysis.

Thank you for your helpful comments. As noted above, we've shifted to a combined Results and Discussion in this revision. We have also added sub-headers to the Results and Discussion.

Reviewer 2: *A more detailed discussion of how the new BRT-based proxy reconstruction compares to that of the traditional MAT approach would be welcome. Does this study indicate that BRT should be widely adopted instead of MAT?*

We have clarified our rationale for preferring BRT over MAT – i.e., mainly cross-validation performance, though acknowledging its limitations when assessing paleo-reconstructions – in an earlier reply (p. 3). A more thorough assessment of reconstruction methods we consider beyond the scope of this manuscript, rather on this we refer to our earlier experiments published in Salonen et al. (2019). Instead we have, following another comment (p. 7), moved the assessment of the MAT reconstructions entirely to the Methods and the supplement.

Reviewer 2: *The authors note that the latest generation of transient climate models do not agree with the reconstructions when it comes to TJul but say that the data-model match is much-improved for the entire growing season heat sum (GDD5) (which is also true for TraCE-21ka). This appears to be truer for the Great Lakes and Northeast region than it is for the Midwest (Figure 2c), which should be acknowledged/explained. Additionally, it would be worthwhile for the authors to describe why it is that the transient simulations produce better agreement for GDD5 than they do for TJul, and why this observation does not hold true for the CESM1 time slice experiments. Finally, is there a reason that the GDD5 values are not included in Supplementary Table 2? (Lines 394 – 402)*

We acknowledge the worse data-model fit for Midwest in the GDD5 paragraph of our revised Results and Discussion.

It is somewhat ironic that our aim was to resolve the model-proxy discrepancy of Holocene water balance (which we do) but now end up with a disagreement in the magnitude of warmer July temperatures. The disagreement of July temperatures between the transient models and our proxy evidence would require more work including in other regions of North America or Eurasia and using other proxies. This is beyond the scope of this study which has a focus on water balance and we prefer to avoid speculation at this point. GDD5 was only a first test to see if the problems are overall to warm summers or only too warm peak summer temperatures in July. GDD5 gives a first indication that peak temperatures are the problem rather than the overall summer warming.

As noted in a reply to Reviewer 1 (p. 11), we have significantly expanded the discussion of the proxy-data match achieved for T_{jul} vs. for GDD5. We have also added the GDD5 values to Supplementary Table 2.

Reviewer 2: *The trend reversal in PC2 after 5ka really stands out (Figure 3, Figure 6). Can the authors clarify what drives PC2? Is it the extent of the LIS up until it is gone at ~5 ka at which point the main driver becomes the east-west moisture contrast? Or is it the east-west moisture contrast across the entirety of the record and PC2 aligns so closely with LIS extent from 11 to 5 ka because the LIS extent controls the east-west moisture contrast? (Line 452 – 454)*

About the PC2 interpretation, based on the spatial pattern in the loadings (Fig. 3c) the early-mid-Holocene fall in PC2 represents the *increase* in moisture contrast as the MW, with a lower baseline moisture level, further dries while the NE starts to get wetter following the early-Holocene drought there. (This the CESM1 simulations can link to the impact of the LIS anticyclone, as we discuss in the manuscript.) The reversal in PC2 trend after 5 ka follows from the stronger increase in water balance in MW compared to NE (seen in Fig. 6a), which now starts to *decrease* the midcontinent-Atlantic moisture contrast.

The interpretation of the PC2 is included in the “Causes of Holocene millennial drought in eastern North America” section of the Results and Discussion. We have clarified the wording in that paragraph.

Reviewer 2: *It is somewhat surprising to the reader that the authors focus the second half of the discussion entirely on the time slice experiments from CESM1. I understand that they focus on CESM1 because they are probing the dynamical relationship between hydroclimate and the LIS, but the authors could do a better job of outlining the rationale for this focus in Lines 456 – 469. Additionally, it would be interesting to see the same dynamical treatment shown in Figures 7 and 8 given to time slices extracted from the transient simulations. These analyses carried out for the transient simulations (individually or as an ensemble) for 8, 7, and/or 6 ka would be useful for further elucidating the influence of the LIS as it disappears. (Lines 490 – 537).*

The CCA analysis is confusing and would benefit from a more fleshed-out description. It would be helpful if the authors explicitly stated what the percentages in the Figure 8 sub-headers mean in terms of the CCA.

The climate models are used to help with the proxy interpretation where two time periods stand out:

1. The early Holocene shows a west-east dipole which has been hypothesized to be caused by the ice sheet. CESM1 is used to evaluate this and confirms the interpretation (wind pattern in Fig. 7, CCA in Fig. 8).
2. The remainder of the Holocene shows long-term transient changes which are now consistent with newer climate model simulations. By merging results and discussions, we can now make these points clearer by using subheadings.

Given the length of the paper and that the remainder of the Holocene is only characterized by consistent long-term trends between models and proxies, we do not see any added value for CCA in this context. The CCA was specifically used to test the role of ice sheets which are absent after ~8 ka BP. The CCA without the role of ice sheets is included in Fig. 8 representing PI.

We have improved the description of the CCA in the revised “Results and Discussion” and in the “Methods”. The explained variances in Fig. 8 indicate how much of the overall variance is explained by the given CCA1 or CCA2 pattern (similar to PCA where the eigenvalue can be used to calculate the explained variances).

Reviewer 2. Methods

What criteria do the authors use to subset the “high-resolution” fossil pollen records?

The criteria were based on the fact that in the Midwest and Northeast, there were subsets of sites which clearly stood out in terms of resolution of pollen data and number of dating (see Supplementary Table 1). The criteria (minimum of 125 pollen samples and 9 dating) are mentioned in Methods. The authors also had good understanding of these sites from prior work, and confidence that these sites have no outstanding issues related to e.g. taphonomy, sedimentation or chronology that would not be obvious based on a simple screening on Neotoma, which increased our confidence in using these thresholds.

Reviewer 2. *The authors should outline which model grid cells are used to calculate mean values for each of the three regions. They could also mark the boundaries of these regions (as defined for the spatial extents in the model output) in Figure 1.*

We have added the coordinates of the boxes in Methods. We now also show these boxes in Fig. 1.

Reviewer 2. *The authors should describe how they calculate the water balance synthesis curves in Figure 6.*

Added some detail in the caption; in general, the methodology is exactly identical with the main results shown in Fig. 2.

Reviewer 2. *Figures*

Figure 4: I'd recommend showing the points for 12, 11, and 9 ka snapshots from CESM1, and potentially doing away with the lines connecting these points, to underscore that these are not transient simulations.

We agree and have split Fig. 4 in two panels (a = CESM1 equilibrium runs for 12, 11 and 9 ka; b = MPI-ESM and EC-Earth transient runs for 8–0 ka) to underline the fundamental difference in the modeling for the two time windows.

Reviewer 2. *Why combine MPI-ESM and EC-Earth for this figure while keeping them separate in Figure 2? Does this model ensemble offer something that the individual models do not? The specifics of how this ensemble is calculated should be described in the text. For example, were the models interpolated to the same spatial resolution prior to averaging? Would it be worthwhile to also include TraCE-21ka in Figure 4?*

In Fig. 2, we consider it necessary to show the raw outputs of all the simulations we have used. However, given the results seen in Fig. 2, namely that the new-generation MPI-ESM and EC-Earth give very similar results, while showing much improved fit with the reconstructions compared to TraCE-21ka, we consider it sensible to only show in Fig. 4 two models (MPI-ESM and EC-Earth) which are consistent with the reconstructions. Further, because the MPI-ESM and EC-Earth results are so similar, we prefer to simplify Fig. 4 by showing their mean instead of two near-overlapping sets of curves, because we consider Fig. 4 somewhat challenging to communicate as-is.

Reviewer 2. *What explains the big difference in sign of the precipitation anomaly between CESM1 at 9 ka and the mean of MPI-ESM1 and EC-Earth at 8 ka? Is this due to differences in the models (CESM1 vs the transient simulations) or interpreted as an actual feature of the climate system?*

As explained above in response to Reviewer 1 (p. 8), the jump is likely a combination of a real change as indicated by the agreement of simulated and reconstructed water balances across models and the switch to different model experiments. As explained further up, the transient models start from equilibrium initial conditions at 8 ka BP and miss out on ocean memory of the system coming out of an ice age.

Reviewer 2. *Can the authors include some metric of uncertainty in this figure, especially for the average of MPI-ESM and EC-Earth.*

We would prefer to leave Fig. 4 unchanged in this respect, as it might look odd to add a standard deviation for the two models in Fig. 4b and have none for CESM1 in Fig. 4a (where there is only one simulation). As the individual model time series are shown in Fig. 2 and have very similar trends, the errors would add little value. It could also make Fig. 4b quite difficult to read as there are already 9 time series shown, and the idea being communicated is already quite complex.

Reviewer 2. *Figure 7: The authors should consider graphically representing areas where the difference between the time slice experiment (12, 11, or 9 ka) and the PI are not significant. This can be done using a Student's T Test and shown on the figures by graying or whiting out the anomaly (usually around the transition from positive to negative).*

The use of significance testing is problematic here given the nature of CESM1 simulations. The SST and hence surface temperature over the ocean is prescribed as climatology and hence has zero variance in time precluding the calculation of significance for temperature. As we do not make any formal comparisons of models or proxies where significance testing would be relevant, we would prefer to use simple anomaly maps to highlight the large-scale changes independently of local grid point significance over land.

Reviewer 2. *Figure 8: I am confused as to what exactly these anomalies are relative to. Can the authors describe how the data in this figure are calculated? Are all these anomalies statistically significant / is there a way to test and communicate this? I understand that CCA1 doesn't always have highest explained variance for P-PET or SLP individually, but is it strange that there are times when the values for both P-PET and SLP are lower than those for CCA2?*

We have now provided a more detailed explanation of CCA in the text. CCA anomalies help identify specific deviations from expected patterns based on the relationships between variables, here local water balance and large-scale SLP. For example, in CCA1, a water balance anomaly of +50 mm indicates that the large-scale SLP linked to the water balance causes the water balance to be 50 mm higher than it would otherwise be. The CCA patterns represent anomaly patterns from the basis used to calculate the EOFs. The interpretation of the according anomalies within the patterns needs to take into account the respective values of the Canonical time coefficients. For instance, a value of 1 in the time coefficients indicates a typical anomaly SLP-local water balance pattern represented in the individual canonical CCA patterns. Higher/lower values in CCA time coefficients scale accordingly and can even lead to inverse anomaly patterns in the presence of negative CCA time coefficients

Reviewer 2: *Lines*

Line 238: What is meant by “a poor spatial coverage of coastal areas in the model.”?

The model resolution even at $\sim 1^\circ$ is still too coarse and does not capture the land-sea distribution of coastal areas well. While proxies from lakes near the coast would still represent conditions over land, the model would average land temperatures with SST grid points making a model vs. proxy comparison inconsistent for any grid point that is not 100% land fraction. To avoid this problem, we set all grids to missing values that include fractional values from SST along the coastline.

Reviewer 2: *Line 353: Add closing parenthesis. Add comma after “In contrast”*

Done.

Reviewer 2: *Line 432: Can the authors briefly elaborate on the “internal variability” that may be contributing to the ~ 200 -year periodicities that are observed in both reconstructions and simulations?*

There is still an ongoing debate about the nature and origin of low-frequency variability. Some control simulations with static forcing of some models do show such low-frequency variability, hence purely internal variability, while others don't. As shown for the EC-Earth model version also used in our study here, internal feedbacks can sustain multi-centennial variability in AMOC (Cao et al. 2023). Externally forced transient simulations including the ones used here typically show variability around 200-year periodicities, consistent with proxies (e.g. Askjær et al. 2022). In that case, multi-centennial variability can be a mixture of forced (e.g. episodes with volcanic eruptions, solar forcing) and unforced internal variations. It is noteworthy that EC-Earth, with only transient forcing changes of radiative forcing (GHG, orbital), shows comparable ~ 200 year variability as MPI-ESM that in addition includes variations in volcanic and solar forcing. We have added the Cao et al. (2023) reference to the manuscript.

Reviewer 2: Line 495: Refer to Figure 8 at end of sentence introducing CCA.

Done.

Reviewer 2: Line 637: Reword the beginning of the sentence about CCSM3.

Done.

Reviewer 3: *This paper presents an impressive synthesis of pollen records and model analyses, leveraging new regression tree methods to attempt to distinguish moisture balance from temperature signals in the eastern US. The boosted regression tree approach appears to provide new insights not available via the common modern analog technique. The authors then compare these results to transient climate model simulations to speculate about the mechanisms responsible for past changes. Overall, I really like this paper, but I think it could benefit from tightening the story and honing the ‘big picture’ message of the work. I think these could be tackled via major revisions. My overall feeling is that the main strength of this paper comes from the interesting new reconstruction. The model analyses are interesting, but also raise many questions and some confusion in those analyses detracts from the overall strength of this work. I would not be afraid to restructure this paper as a ‘data driven’ paper.*

Below are some overall points to consider as well as detailed comments on the text.

First, I think the organization of the paper would benefit from a combined results and discussion section rather than having these sections distinct. As it stands, there are topics that should traditionally be included in the discussion in the Results, and the organization gets a little bit messy. Especially for a short format paper, combining these would be valuable.

Thank you for these helpful comments and for your positive feedback. As noted above, we’ve shifted to a combined Results and Discussion (with subheadings).

We note that the main motivation to conduct this study was twofold: (1) opportunistically, the availability of new proxy methods and new model simulations and (2) the pending mismatch between climate simulations and proxy evidence regarding water balance trends across the Holocene. We hence re-investigate the issue by employing a range of classical and new proxy methods which are compared with new climate model simulations to resolve the issue. As shown in Fig. 2a, the mismatch is now mainly attributable to the older model simulations with TraCE-21k, which is resolved by the newer model simulations rather than the proxy reconstruction. We hope this is clearer in our revised version.

Reviewer 3: *Second, the overall conclusion of the paper could be stronger. The abstract reads more like a literature review rather than the abstract of a short format paper with a ‘punchy’ conclusion, and a similar comment could be applied to the paper’s discussion. My personal thought is it might be better to focus either on the impact of long-term forcing on the PCA/water balance reconstruction, or on the question of periodicities. As it stands, I don’t know that the model analysis is ‘enough’ to support the conclusion that summer insolation and the Laurentide are driving the patterns you see. But, it is worth noting that Trace21ka has single forcing simulations (e.g. with ONLY orbital changes, or with ONLY ice volume). : <https://www.earthsystemgrid.org/project/trace.html>*

It is also worth taking a step back and starting with evaluating the PI model simulations’ climatology. Which ones are too rainy over your regions of interest? Too hot? Are the winds too strong relative to observations? This might provide some additional insights about why model performance varies. It would also be good to make a table as to what

forcing datasets were used in each set of transient runs - the change in different generations of ice sheet reconstructions (e.g. 5G, 6G, etc) will affect simulations even of the same model.

Finally, there are other ‘data’ products out there for deglacial climate variables (see Osman et al 2022 in Nature for the data-assimilation approach). My sense is that that product performs well for temperature but might not perform well for hydroclimate-related variables. It would be worth comparing your reconstructions (real world data! So more likely to represent the truth) to the DA product for P and P/ET during the deglaciation. It might help emphasize the innovative nature of the reconstructions that you’ve produced.

As mentioned above about the scope of the paper, our motivation for the paper was both the availability of a new robust pollen–water balance calibration as well as the increasing body of well-dated, high-resolution fossil pollen sequences. This paper was then deliberately written to approach the datasets from the various angles allowed by the data properties. We acknowledge that the resulting paper is somewhat complex and our decision to submit to *Nature Communications*, with its fairly relaxed manuscript format, was deliberate.

We acknowledge the problem with our initial submission’s long and convoluted Discussion spanning many topics. Our revised manuscript’s Results and Discussion is now divided in thematic subsections, and also includes some new introductory and concluding sentences for the thematic sections, which we hope make our main conclusions stand out better.

Unless there is a specific reason, it is generally not common to re-evaluate the performance of already tested state-of-the-art climate models regarding their systematic biases. Instead, these systematic model biases are usually removed by, e.g., subtracting the PI simulations from past (or future) simulations. This standard procedure is applied here so that the focus is only on relative changes over time. It should also be noted that biases in the PI do not necessarily say much about biases in, e.g. CESM1, for periods with fundamentally different ice sheets and land-sea distributions, as is the case for the early Holocene. We hence do not see merit in speculating about how present-day biases – which we removed – might or might not influence transient changes in the past. The model evaluation of these changes is here based on the agreement with proxy data.

We agree with the reviewer that single forcing runs like available for TraCE-21k would be an interesting addition to explore temperature and water balance changes further. This is however beyond the scope of this already long study. In addition, the fundamental mismatch of the full-forcing TraCE-1k result with proxies and the new transient simulations in this study (Fig. 2) would, a priori, raise questions about its validity.

Reviewer 3: Introduction

Line 126-131: Instead of just reviewing the methods and approaches you used in the paper, provide a signpost of the overall conclusion of the paper.

With the revised structure of the Results and Discussion, with six thematically titled subsections, combined with the Abstract, we hope our main conclusions are now easy to identify in the paper. We’d prefer to not reiterate the findings and the end of Introduction because they are, due to the nature of the paper, somewhat wordy to express.

We have, instead, edited the first paragraph of Introduction for brevity, including removing mention of the MAT test which is now out of the main manuscript text. The paragraph should now read more as a concise list of the manuscript’s goals, and lead logically to the immediately following Results and Discussion subsections.

Reviewer 3: Results:

Figure 2: Minor point, but I find it hard in some panels to distinguish the lines for EC-Earth and Trace 21ka. Make sure you check colorblindness compatibility of the figure, and perhaps use line weights/dash patterns to increase understandability.

We have shifted all figures which show either proxy data and transient model simulations to use the Bang Wong color blind friendly palette (<https://davidmathlogic.com/colorblind>). The palettes in the equilibrium model simulated fields in Figs. 7 and 8 are unchanged, as they already avoid using red and use white to separate negative and positive values.

Reviewer 3: *Line 183: This applies to the paper overall, but it is worth defining what you mean by ‘drought’ - for paleo records like this, we tend to define drought as just below average P or P-E, while the term can mean something very different in the modern literature (e.g. below a certain percentile of precipitation).*

We define “drought” here as annual water balance being under the preindustrial value. This is articulated in the first sentence of our revised Results and Discussion.

Reviewer 3: *Line 196: I think the PCA presents one of the most interesting results in this paper, and it is worth really strongly pinning down what the driver of this pattern is. I am not sure I understand why PC2 is Laurentide deglaciation if it goes ‘up’ again after 5 ka?*

See earlier reply about PC2 interpretation (p. 16).

Reviewer 3: *Line 225: I am not sure I agree that there is a good comparison with the evolution of water balance between the models and the data. CESM1 seems to overlap the water balance reconstructions, but seems to be very off in terms of GDD so this to me suggests that it may be getting the ‘right’ answer in terms of hydroclimate change for the wrong reasons. I am therefore not sure how much to trust the CCA. The logic behind relying on CESM1 for interpretation needs to be clearer. An alternate/complementary analysis might involve comparing Trace to MPI and EC Earth for an interval like 7 ka, when the former is not dry enough and the latter two are too dry. Difference fields between these runs might reveal why the models differ in skill at simulating deglacial climate change. See comments above about some basic diagnostic analyses as well.*

By design, the water balance may include compensating effects between PET (which is driven by summer temperatures and incorporating the season length above freezing) and annual precipitation. The relatively good agreement of simulated vs. reconstructions, i.e. regarding the consistency of trends, is a clear improvement to previous studies. The additional comparison of reconstructed vs. simulated GDD5, which characterizes the overall warm season instead of peak July temperatures (now Supplementary Fig. 1) indicates a better agreement of GDD5 between proxies and MPI-ESM and EC-Earth, suggesting that the problem is rather in temperature and hence PET than the water balance itself. For CESM1, the problem is more the seasonal GDD5 rather than July, when comparing to proxies. So in both cases, we would rather trust the water balance than the details of peak July temperature. Hence, we’re confident that the CCA for water balance is reasonable given the proxy-model agreement for water balance. The problem is rather in temperature (seasonal vs. peak July disagreement).

As noted in a reply to Reviewer 1 (p. 11), we have significantly expanded the paragraph discussing the significance of the GDD5 comparison between proxies and models.

Reviewer 3: *Figure 6: In panel d, if this mode reflects Laurentide deglaciation, why does it go back ‘up’ slightly after 5 ka?*

See earlier reply about PC2 interpretation (p. 16).

Reviewer 3: *Line 305: I am not sure if the section on periodicities is a good idea for this paper. It seems unrelated to the overall story of the paper, and could be a separate, more robust analysis in and of itself.*

See earlier reply about our rationale for the scope of the paper (p. 21).

Reviewer 3: Discussion:

Suggest combining with results section, much of this discussion could be integrated through the prior section

As noted above, we have done this.

Reviewer 3: Line 394: *See comments above about model-data comparison.*

We refer back to that reply (p. 22).

Reviewer 3: Conclusions:

The concluding paragraph of your paper focuses on the MPI, EC Earth, and new reconstructions. If this is the focus, it would be important to then focus on these model simulations in Figures 7 and 8.

We have added subheadings in the Results and Discussion which should clarify the reason for the shift from CESM1 to MPI-ESM and EC-Earth, as we no longer deal with the period affected by the ice sheet (which we study using the CESM1 runs).

Reviewer 3: *I would also check about the length limits for this journal - it seems like there are a lot of figures, and the text is quite long for a short format journal.*

We're currently within the length limits, which for *Nature Communications* are in fact somewhat relaxed (5000 words excluding Methods, and 10 display items).

In addition to the changes described above, we have made minor edits for grammar and clarity. Also, we note that we plan to improve the quality of the Supplementary Figures 1–10 and 12–13 for the final revisions before publication. This will include a better resolution, and a taller vertical scaling of the reconstruction panels in Supplementary Figures 1–10, making them easier to read. We've had technical issues exporting from R the dotted and dash-dot lines which are used in these figures for the individual fossil sites. We'll sort this out for the final revisions.

Finally, we wish to again thank the reviewers once more for their thorough and thoughtful comments.

Respectfully,

Helsinki, 6 December 2024

Dr. J. Sakari Salonen, on behalf of the co-authors

E-mail: sakari.salonen@helsinki.fi

References

- Askjær TG, Zhang Q, Schenk F, Ljungqvist FC, Lu Z, Brierley CM, Hopcroft PO, Jungclaus J, Shi X, Lohmann G, Sun W, Liu J, Braconnot P, Otto-Bliesner BL, Wu Z, Yin Q, Kang Y, Yang H (2022) Multi-Centennial Holocene Climate Variability in Proxy Records and Transient Model Simulations. *Quaternary Science Reviews* 296:107801. <https://doi.org/10.1016/j.quascirev.2022.107801>
- Cao N, Zhang Q, Power KE, Schenk F, Wyser K, Yang H (2023) The Role of Internal Feedbacks in Sustaining Multi-Centennial Variability of the Atlantic Meridional Overturning Circulation Revealed by Ec-Earth3-Lr Simulations. *Earth and Planetary Science Letters* 621:118372. <https://doi.org/10.1016/j.epsl.2023.118372>

- Hébert R, Herzschuh U, Laepple T (2022) Millennial-Scale Climate Variability over Land Overprinted by Ocean Temperature Fluctuations. *Nature Geoscience* 15:899-905. <https://doi.org/10.1038/s41561-022-01056-4>
- Kuang X, Schenk F, Smittenberg R, Hällberg P, Zhang Q (2021) Seasonal Evolution Differences of East Asian Summer Monsoon Precipitation between Bølling-Allerød and Younger Dryas Periods. *Climatic Change* 165:19. <https://doi.org/10.1007/s10584-021-03025-z>
- Salonen JS, Korpela M, Williams JW, Luoto M (2019) Machine-Learning Based Reconstructions of Primary and Secondary Climate Variables from North American and European Fossil Pollen Data. *Scientific Reports* 9:15805. <https://doi.org/10.1038/s41598-019-52293-4>
- Schenk F, Väiliranta M, Muschitiello F, Tarasov L, Heikkilä M, Björck S, Brandefelt J, Johansson AV, Näslund J-O, Wohlfarth B (2018) Warm Summers During the Younger Dryas Cold Reversal. *Nature Communications* 9:1634. <https://doi.org/10.1038/s41467-018-04071-5>

PATTERNS AND DRIVERS OF HOLOCENE MOISTURE VARIABILITY IN MID-LATITUDE EASTERN NORTH AMERICA

Response to Referees

We quote below the reviewers' full comments and describe the changes done to address each point. We thank all the reviewers for their helpful comments.

Reviewer #1: *Dear Editor,*

I am generally satisfied with the authors' responses and have no major reservations regarding the manuscript. They have addressed my comments in many cases and have carefully justified their choices in others. I appreciate their revisions, particularly the improved methodology for stacking and the clarifications of other methodological aspects.

One aspect that still leaves me somewhat uncertain is the analysis of periodicities. I believe the current framing places too much emphasis on the least informative aspect—wiggles in the wavelet spectra that, after appropriate bandpass filtering, are interpreted as significant periodicities. That said, I appreciate that the authors have tested the sensitivity of their periodicity estimates by varying the null hypothesis, and the analysis is still informative of the distribution of variability across frequencies.

If there are no restrictions on adding new supplementary figures, I would still recommend including the mean wavelet plots used for the periodicity estimates, as these would provide valuable insights. Specifically, plotting the individual wavelet spectra for all sites within a cluster together would allow the reader to see the "periodicities" clustering and, importantly, the overall shape of the wavelet spectra. Alternatively, the mean wavelet spectra could be shown for each region (i.e. a stack of the individual spectra, averaging in the spectral rather than the time domain) with shading to indicate the spread among individual records.

Overall, I see no reason to further delay the publication of the manuscript. Best regards

We have added the mean power plots of all wavelet spectra (shown in Supplementary Figures 18–25) in a new Supplementary Figure 26. This figure is cited, like Supplementary Figures 18–25, in the caption to Figure 6 which presents the wavelet results in the main manuscript.

Reviewer #2: *The authors have submitted a response letter and updated version of the manuscript that effectively addresses the comments raised by each of the three reviewers. The reorganization of the Results/Discussion section in particular has strengthened the manuscript. I recommend that the updated version of the manuscript be published and thank the authors for their detailed work to address reviewer comments.*

We thank the reviewer once more for their helpful comments on this manuscript.

Reviewer #3: *The authors have largely addressed my comments and those of the other reviewers. I could use some extra clarification on this text near line 428:*

"This spatiotemporal pattern has been hypothesized to be explained by LIS deglaciation, and a following waning of the glacial anticyclone and an increased influence of the Bermuda subtropical high, leading to a rerouting of the northward moisture advection towards the NE region and away from the mid-continent"

I'd like a bit more detail - this sentence seems to suggest that after the Laurentide is gone,

the upwards trend of PC2 post 5ka could reflect shifts in the strength of the North Atlantic Subtropical High? it would be nice to propose that explicitly in the text.

The rotational component of the western part of the Bermuda high generally helps to advect air and moisture from the (sub-)tropics northward over the US. This northward advection would be reduced while the LIS reaches far south. The role of the Bermuda subtropical high was mentioned as part of the original hypothesis in the literature but doesn't imply a change in the high itself. It is rather the retreat of the ice sheets which allow the flow and moisture transport to move further north and by doing so increases the influence of the Bermuda high.

With the results presented in this paper, we have no means to infer potential changes in the Bermuda high itself in terms of proxy data, and so we find it difficult to address this issue in the text. The CESM1 simulations do not show any change in the wind or SLP pattern of the Bermuda high itself for the period 12, 11 and 9 ka (Fig. 7a,d,g).

Reviewer #3: *Given the revised structure of the paper, it would be nice to have a conclusion section that summarizes the overall argument of the paper and a few thoughts on future directions. The last paragraph could be slightly expanded and serve as your Conclusion section.*

We have edited the last subsection of Discussion to start by recapping the fact that the early Holocene hydroclimate features showed a significant impact of the Laurentide Ice Sheet – and then transition to discussing the temperature and evapotranspiration dominated regime of the mid-late Holocene. With this, we feel the last subsection of Discussion adequately summarizes the main features of the entire Holocene, while concluding with a forward-looking perspective.

We're open to adding a separate Conclusions section, perhaps as bullet points, if the editor feels this would be helpful. But on balance we'd favour going without, as we're already at the word limit.

Reviewer #3: *For the abstract, while it is true that the midHolocene features evaporatively-driven drying similar to the future, it does not necessarily follow that they are analogs, which to me implies a symmetry of underlying mechanisms (and the radiative forcing of CO2 is different in character than seasonal changes in insolation). Rather, it seems more fair to say that similar processes are at play. I would revise the abstract accordingly.*

Agreed; we have edited the Abstract to state that "...the mid-Holocene multi-millennial drought was driven by similar processes compared to the drying trajectory projected..."

In addition to the changes described above, we have done the following changes

- Fixed two cases of "analogue" as "analog".
- We have replotted Supplementary Figures 1-10 (paleoclimate reconstructions including results for individual fossil sites) and Supplementary Figures 12 and 13 (diagnostic plots of most important predictor taxa in the calibration models) to make them more readable. This does not change the data shown in the figures. The vertical scaling of the reconstruction panels in Supplementary Figures 1–10 has been increased, and Supplementary Figures 12 and 13 now use correct line symbology for the fossil sites.
- We have added a Data Availability statement in the manuscript. We have uploaded the Supplementary Data file on Figshare (with a reserved DOI but not yet published).

Finally, we wish to once more warmly thank the reviewers for their thoughtful comments during this review process, which have greatly helped us improve the manuscript.

Respectfully,

Helsinki, 25 February 2025

Dr. J. Sakari Salonen, on behalf of the co-authors

E-mail: sakari.salonen@helsinki.fi